# CXCL12-induced rescue of cortical dendritic spines and cognitive flexibility

Lindsay K Festa[1,2†], Elena Irollo[1], Brian J Platt[1], Yuzen Tian[1], Stan Floresco[3], Olimpia Meucci[1,2,4]*

[1]Department of Pharmacology and Physiology, Drexel University College of Medicine, Philadelphia, United States; [2]Center of Neuroimmunology and CNS Therapeutics, Institute of Molecular Medicine and Infectious Diseases, Drexel University College of Medicine, Philadelphia, United States; [3]Department of Psychology, University of British Columbia, Vancouver, Canada; [4]Department of Microbiology and Immunology, Drexel University College of Medicine, Philadelphia, United States

*For correspondence:
om29@drexel.edu

Present address: [†]Department of Basic and Translational Sciences, University of Pennsylvania School of Dental Medicine, Philadelphia, United States

Competing interests: The authors declare that no competing interests exist.

**Abstract** Synaptodendritic pruning is a common cause of cognitive decline in neurological disorders, including HIV-associated neurocognitive disorders (HAND). HAND persists in treated patients as a result of chronic inflammation and low-level expression of viral proteins, though the mechanisms involved in synaptic damage are unclear. Here, we report that the chemokine CXCL12 recoups both cognitive performance and synaptodendritic health in a rodent model of HAND, which recapitulates the neuroinflammatory state of virally controlled individuals and the associated structural/functional deficiencies. CXCL12 preferentially regulates plastic thin spines on layer II/III pyramidal neurons of the medial prefrontal cortex via CXCR4-dependent stimulation of the Rac1/PAK actin polymerization pathway, leading to increased spine density and improved flexible behavior. Our studies unveil a critical role of CXCL12/CXCR4 signaling in spine dynamics and cognitive flexibility, suggesting that HAND - or other diseases driven by spine loss - may be reversible and upturned by targeting Rac1-dependent processes in cortical neurons.

## Introduction

The neurological complications of human immunodeficiency virus 1 (HIV-1) infection, collectively known as HIV-associated neurocognitive disorders (HAND), remain an important and unmet clinical need (*Heaton et al., 2010*). While the introduction of combination antiretroviral therapy (ART) has significantly reduced the severity of neurological impairments in HIV+ individuals, approximately 30–50% of infected patients will develop some form of neurocognitive dysfunction (*Saylor et al., 2016*). Additionally, these impairments continue to be important determinants of quality of life, as well as disease progression (*Simioni et al., 2010*; *Tozzi et al., 2007*).

The neuropathology of HAND is complex and has shifted dramatically since the implementation of ART (*McArthur et al., 2010*); however, the presence of chronic, low-level inflammation and viral neurotoxins disrupt neuroprotective and bioenergetic mechanisms that converge on aberrant synaptodendritic pruning and functional alterations in neurons (*González-Scarano and Martín-García, 2005*; *Raybuck et al., 2017*; *Sanchez et al., 2016*; *Sanna et al., 2017*; *Saylor et al., 2016*; *Stern et al., 2018*). Developing novel adjuvant therapies that correct these dendritic alterations may be an effective strategy to help neurons face the multiple chronic insults associated with HIV infection and rescue cognitive function in these patients. A reduction in the number of dendritic spines, the main sites of excitatory neurotransmission, in the frontal cortex of HIV+ patients is significantly correlated with neurocognitive impairment (*HNRC Group et al., 1999*; *Masliah et al., 1997*). This finding has been recapitulated in animal models of HAND, including the non-infectious HIV-1

transgenic (HIV-Tg) rat (*Festa et al., 2015*; *Reid et al., 2001*). HIV-Tg rats are being increasingly used in HAND research, as they recapitulate key neurochemical, genetic, and behavioral features of the human pathology as seen in treated patients, such as undetectable viral replication, reduction of post-synaptic markers, neuroinflammation, changes in gene expression, and behavioral deficits (*Festa et al., 2015*; *Homji et al., 2012*; *June et al., 2009*; *McLaurin et al., 2017*; *Reid et al., 2001*; *Repunte-Canonigo et al., 2014*; *Royal et al., 2012*; *Vigorito et al., 2015*; *Wayman et al., 2016*). In these animals, as in patients, viral proteins are expressed at low levels in the brain - including in the prefrontal cortex, an area involved in higher cognitive functions and neuroHIV (*Ann et al., 2016*).

The chemokine CXCL12 and its main signaling receptor CXCR4 regulate several critical steps of CNS development, including neuronal survival and neuronal-glial communication (*Bhattacharyya et al., 2008*; *Guyon and Nahon, 2007*; *Khan et al., 2004*; *Nicolai et al., 2010*). Our previous findings in differentiated central neurons suggested that the chemokine could also modulate spine morphology, an important factor for input-specific alterations in synaptic strength (*Yuste, 2011*), particularly in the mature brain. Dendritic protrusions are often grouped into four major classes: mushroom, thin, and stubby spines and filopodia (*Peters and Kaiserman-Abramof, 1970*). Mushroom and thin spines are classically considered mature structures, while stubby spines and filopodia are thought of as immature or spine precursors, respectively (*Bourne and Harris, 2007*).

Here, we report that intracebroventricular (ICV) administration of CXCL12 completely rescues dendritic spine loss and cognitive dysfunction in the HIV-Tg rat. The chemokine specifically increased the number of thin spines, which are associated with learning and plasticity, in layer II/III pyramidal neurons of the medial prefrontal cortex (mPFC). In the same animals, upregulation of spine density by CXCL12 correlates with an improvement in cognitive flexibility, a task mediated by the mPFC. Our mechanistic studies demonstrate the essential role of the Rac1/PAK pathway in mediating the effects of CXCL12 on dendritic spines and cognition. Overall, this study provides evidence of complete restoration of structural and functional deficits in a model of HAND, reveals a novel fundamental role of CXCL12 in the mature brain, and identifies the molecular pathway responsible for CXCL12-mediated restoration of dendritic spines and cognitive performance.

## Results

### Attentional set-shifting is impaired in adult male HIV-Tg rats

We have previously reported reductions in overall dendritic spine density in layer II/III pyramidal neurons in the mPFC in HIV-Tg rats (*Festa et al., 2015*); however, it is currently unknown if these pathophysiological alterations result in discernable cognitive dysfunction. We assessed cognitive flexibility in adult male WT and HIV-Tg rats utilizing an automated three-phase lever pressing paradigm (*Brady and Floresco, 2015*). In the first two phases of the task (position discrimination and position reversal), there was a slight but significant difference between WT and Tg rats for position discrimination ($30.17 \pm 2.212$ TTC for WT compared to $52.50 \pm 11.90$ TTC for Tg; *Figure 1A*) and none observed for position reversal ($81.17 \pm 18.11$ TTC for WT compared to $64.50 \pm 4.19$ TTC for Tg; *Figure 1B*). In the shift to cue phase, however, Tg rats performed significantly worse than control animals (*Figure 1C*). Only four out of six Tg animals reached criterion by day 10 of the set-shifting phase and those who did complete the task took significantly longer to reach criterion ($949.5 \pm 146.3$ TTC compared to $350.8 \pm 84$ TTC for WT). Additionally, a linear regression analysis on the cumulative proportion of animals reaching criterion over sessions revealed that a different curve was best fit for each group ($F_{(1,11)}=15.94$, p=0.0021) further illustrating the gap between the groups in the number of sessions needed to reach criterion (*Figure 1D*).

While we have previously reported changes in dendritic spine numbers in the Tg rat (*Festa et al., 2015*), it is unclear whether subtle alterations in spine morphology, which have implications for input-specific changes in synaptic strength, also occur. From the same set of animals used in the above-mentioned behavioral study, we assessed dendritic spine density and morphology in layer II/III pyramidal neurons in the prelimbic (PrL) region of the mPFC. Compared with control animals, Tg rats had significantly reduced dendritic spine density, as we previously observed in older animals (*Festa et al., 2015*) (*Figure 2A*). In addition to spine density, subtle spine morphological changes were also present, whereby the density of thin (*Figure 2B*) and mushroom spines was reduced

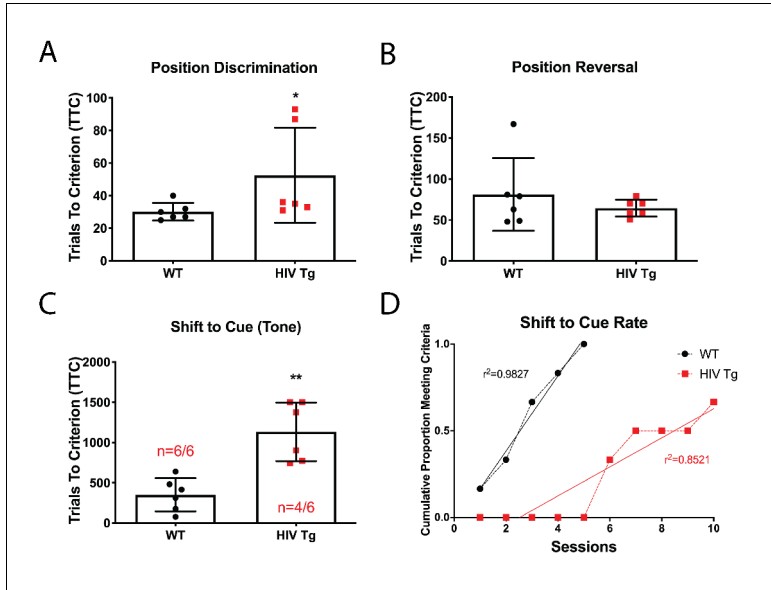

**Figure 1.** Male adult HIV-Tg rats are deficient in strategy shifting and initial rule discrimination, but not reversal learning. (**A**) HIV-Tg rats performed worse on the position discrimination task compared to age- and sex-matched WT animals. N = 6 animals/group, *p<0.05. (**B**) No significant differences were observed between WT and HIV-Tg rats on position reversal. N = 6 animals/group (**C**) Fewer HIV-Tg rats reached criterion on the shift to cue phase and took significantly greater number of trials to reach criterion. N = 6 animals/group, **p<0.01. (**D**) HIV-Tg rats took significantly longer to reach criterion as assessed via linear regression. N = 6 animals/group, ($F_{(1,11)}$=15.94, p=0.0021).

The online version of this article includes the following source data for figure 1:

**Source data 1.** HIV-Tg behavior raw data and statistical analysis.

(*Figure 2C*), while stubby spines were upregulated (*Figure 2D*). This demonstrates a shift in spine morphology from mature (mushroom and thin) to immature (stubby), which is expected to alter cognition. Indeed, spine density in the mPFC was negatively correlated with performance to the shift to cue phase (Pearson's r = −0.7368, p=0.0151, α = 0.05; *Figure 2E*) and this relationship became stronger when only thin spine density was considered (Pearson's r = −0.7920, p=0.0063, α = 0.05; *Figure 2F*). On the contrary, spine density in the mPFC was not correlated with performance on position discrimination (Pearson's r = −0.4548, p=0.1374, α = 0.05) or position reversal (Pearson's r = 0.4862, p=0.1090, α = 0.05; *Figure 2—figure supplement 1*).

## CXCL12 treatment enhances cognitive flexibility in HIV-Tg rats

Based on the above findings and our previous work on neuronal CXCR4 signaling (*Nicolai et al., 2010*; *Pitcher et al., 2014*), we hypothesized that exogenous administration of CXCL12 could alleviate structural and cognitive deficits in Tg rodents. Tg rats were implanted with a unilateral cannula targeted to the lateral ventricle and allowed to recover for seven days prior to initiation of once-a-day infusions with vehicle (0.1% BSA in PBS, 5 µL total volume) or CXCL12 (5 ng/µL, 5 µL total volume) throughout the duration of the three-phase behavioral task (*Figure 3—figure supplement 1*).

Both groups performed similarly across the first two phases (position discrimination and position reversal) of the behavioral task, with animals reaching criterion in approximately one day (*Figure 3—figure supplement 2*). In the shift to cue task, CXCL12-treated rats took significantly fewer trials to reach criterion than vehicle-treated counterparts (for animals who reached criterion: 701.3 ± 97.26 TTC for CXCL12 compared to 1191 ± 114.5 TTC for vehicle). Additionally, a greater number of rats given CXCL12 (n = 7 out of 10) completed the task compared to vehicle-treated animals (n = 2 out of 9; p=0.03; *Figure 3A*). A linear regression analysis on the cumulative proportion of animals reaching criterion over the ten sessions demonstrated that the CXCL12-treated rats completed the task at a significantly faster rate, $F_{(1,16)}$=44.3781, p<0.001 (*Figure 3B*), further emphasizing that CXCL12 rescues cognitive flexibility in this animal model, even after deficits are already present. Parallel studies

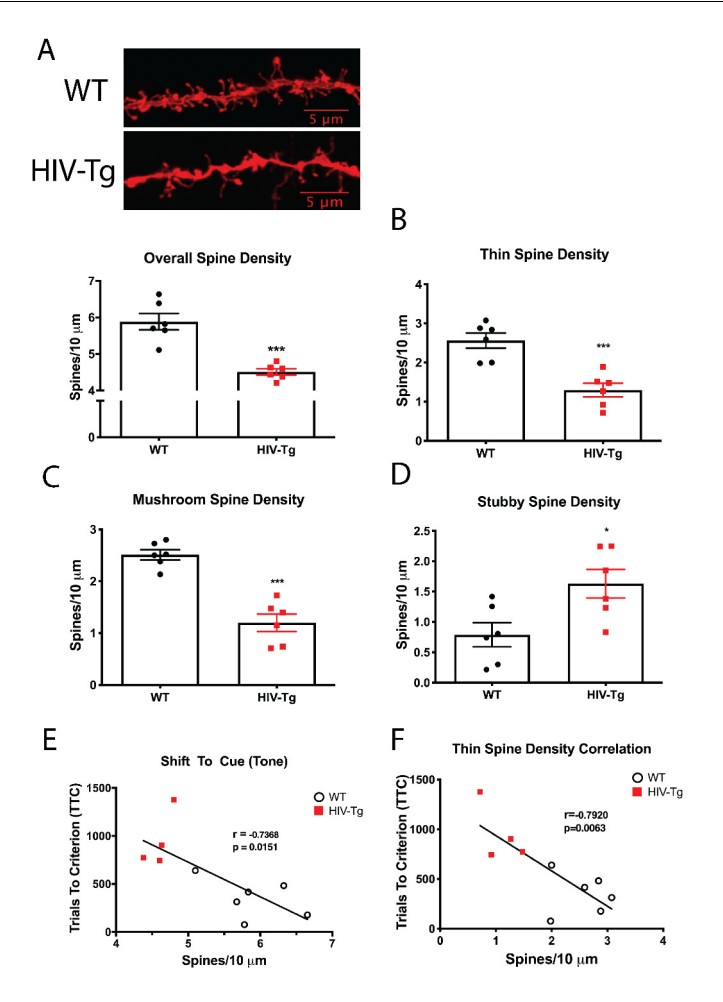

**Figure 2.** Layer II/III pyramidal neurons in the PrL region of the mPFC of HIV-Tg rats have alterations in spine density and morphology. (A) Dendritic spine density and morphology were assessed using Neurolucida 360 software. Male HIV-Tg rats had a reduction in overall dendritic spine density in layer II/III pyramidal neurons in the mPFC. N = 6 animals/group, 8 dendrites measured for each animal and averaged into single data point. ***p<0.001. (B) On the same set of dendrites, thin spine density was significantly decreased in male HIV-Tg rats compared to sex-matched WT controls. N = 6 animals/group. ***p<0.001. (C) As seen with overall and thin spine density, HIV-Tg rats had reduced mushroom spine density compared to WT controls. N = 6 animals/group. ***p<0.001. (D) In contrast, stubby spine density was significantly elevated in male HIV-Tg rats. N = 6 animals/group. *p<0.05. (E) Across both groups, overall dendritic spine density in layer II/III pyramidal neurons in the mPFC were negatively associated with trials to criterion in the shift to cue phase. N = 12 animals; Pearson's r = −0.7368, p=0.0151. (F) When only thin spine density was considered, the relationship with shift to cue trials to criterion became even stronger. N = 12 animals; Pearson's r = −0.7920, p=0.0063.

The online version of this article includes the following source data and figure supplement(s) for figure 2:

**Source data 1.** HIV-Tg dendritic spines raw data and statistical analysis.

**Figure supplement 1.** There was no relationship between overall dendritic spine density and the first two phases of the behavioral task.

in male WT rats (*Figure 3—figure supplement 3*) showed that vehicle- and CXCL12-treated WT rodents performed at a similar level across all three phases of the task, indicating that long term exposure to CXCL12 does not result in any deleterious effects on cognition as measured by the attentional set-shifting task.

In this same set of animals, we evaluated spine density and morphology in layer II/II pyramidal neurons of the mPFC. Once-a-day infusion with CXCL12 significantly upregulated overall dendritic

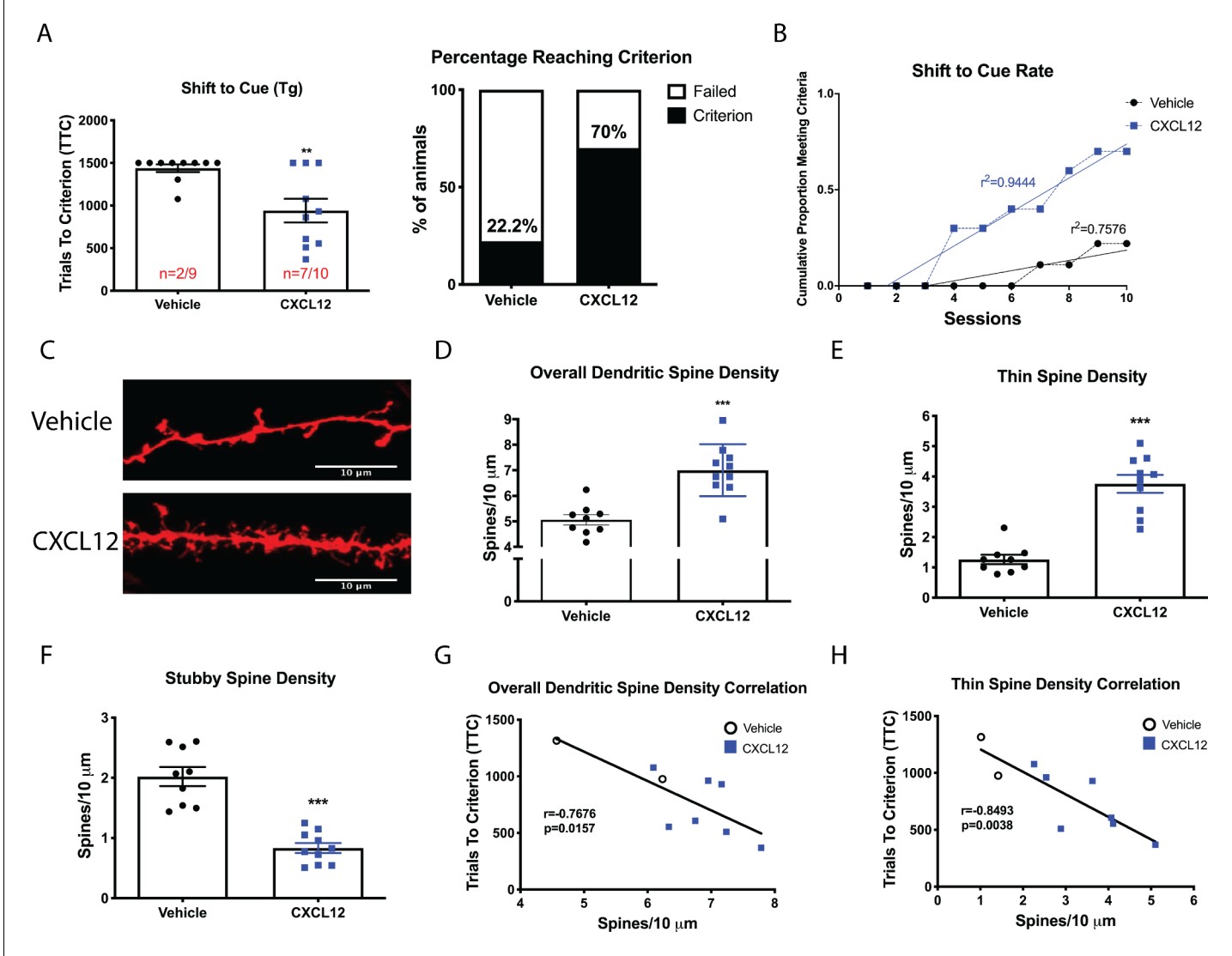

**Figure 3.** Once daily CXCL12 treatment significantly enhanced set-shifting abilities of HIV-Tg rats and completely restored dendritic spine density and morphology in the mPFC. (**A**) A greater percentage of CXCL12-treated rats reached criterion on the shift to cue phase and also took significantly fewer trials to complete the task. N = 9 animals/vehicle and 10 animals/CXCL12, p=0.03. (**B**) The rates at which the groups reached criterion were significantly different, with CXCL12-treated animals learning at a faster rate. N = 9 animals/vehicle and 10 animals/CXCL12, $F_{(1,16)}$=44.3781, p<0.001. (**C**) Representative image of DiOlistically labeled mPFC slices from HIV-Tg rats treated with either vehicle or CXCL12. (**D**) Adult male HIV-Tg rats treated with CXCL12 had significantly higher overall dendritic spine density in layer II/III pyramidal neurons of the mPFC. N = 9 animals/vehicle and 10 animals/ CXCL12, 8 dendrites measured for each animal and averaged into single data point, ***p<0.001 (**E**) Further analysis on spine morphology revealed a specific increase in the number of thin spines. N = 9 animals/vehicle and 10 animals/CXCL12; ***p<0.001. (**F**) CXCL12 treatment resulted in a significant reduction in stubby spine density in HIV-Tg rats. N = 9 animals/vehicle and 10 animals/CXCL12; ***p<0.001 (**G**) As previously observed, overall dendritic spine density was significantly negatively associated with trials to criterion on the set-shifting phase. N = 19 animals, Pearson's r = −0.7676, p=0.0157. (**H**) This relationship became even stronger when only thin spines were correlated with the number of trials required to complete the task. N = 19 animals, Pearson's r = −0.8493, p=0.0038.

The online version of this article includes the following source data and figure supplement(s) for figure 3:

**Source data 1.** WT and HIV-Tg ±CXCL12 raw data and statistical analysis.

**Figure supplement 1.** Experimental timeline for behavior and ICV infusions.

**Figure supplement 2.** Vehicle and CXCL12-treated HIV-Tg rats performed similarly during position discrimination and position reversal.

**Figure supplement 3.** Vehicle and CXCL12-treated male WT rats performed equally in the attentional set-shifting task.

**Figure supplement 4.** CXCL12-treated HIV-Tg and vehicle-treated WT rats had no difference in overall dendritic spine density.

**Figure supplement 5.** CXCL12-treated WT rats display higher overall and thin dendritic spine density in the mPFC.

spine density (*Figure 3C,D*), as well as the number of thin spines (*Figure 3E*), while also decreasing stubby spine density (*Figure 3F*). There were no observed changes in mushroom spines or filopodia. Strikingly, overall spine density in CXCL12-treated Tg rats was not significantly different from WT rats treated with vehicle, consistent with the complete recovery of behavior we observed in these rodents (*Figure 3—figure supplement 4*). We observed similar effects on spine density and morphology in WT rats (*Figure 3—figure supplement 5*), despite no perceived alterations in behavior, most likely reflecting a ceiling effect. Overall spine density in the mPFC was negatively correlated with trials to criterion in the shift to cue phase (Pearson's $r = -0.7676$, p=0.0157, $\alpha = 0.05$; *Figure 3G*), while thin spine density demonstrated an even stronger relationship with performance (Pearson's $r = -0.8493$, p=0.0038, $\alpha = 0.05$; *Figure 3H*). Taken together, this is the first study to demonstrate complete recovery of both dendritic spine density and cognitive performance in the HIV-Tg rat, pointing to the potential therapeutic value of enhanced neuroprotective CXCR4 signaling.

## CXCL12 activates Rac1 in a CXCR4- and G$\alpha$i-dependent manner in cortical neurons

The mechanism by which CXCL12/CXCR4 signaling modulates dendritic spine density and morphology is presently unknown. There is substantial evidence in non-neuronal cells that CXCL12 can activate the small Rho GTPase, Ras-related C3 botulinum toxin substrate 1 (Rac1) that acts as a critical regulator of actin polymerization and stabilization. When Rac1 is in its active form, it is bound to GTP, which allows it to physically interact with the serine/threonine kinase PAK1. Thus, we utilized a Rac1 activity assay that uses PAK1 agarose beads to specifically pull down Rac1 bound to GTP. Cultured neurons (21 DIV) were treated with CXCL12 (20 nM) and lysates were subjected to pulldown to isolate Rac1-GTP. In line with previous reports in non-neuronal cells, CXCL12 significantly upregulates levels of Rac1-GTP at both 5 and 15 min with a return to basal levels by 30 min (*Figure 4A*). Furthermore, there were no alterations in the total levels of Rac1, demonstrating that CXCL12 is altering the amount bound to GTP.

CXCR4 and its coupling to pertussis-toxin (PTX) sensitive G$\alpha$i proteins is essential for the ability of CXCL12 to upregulate dendritic spine density (*Pitcher et al., 2014*). Pretreatment with either the specific CXCR4 inhibitor (AMD3100; 100 ng/mL, 20 min) or PTX (100 ng/mL, 18 hr) completely blocks the ability of CXCL12 to activate Rac1 in cortical neurons (*Figure 4B and C*). These results, as observed with dendritic spine density (*Pitcher et al., 2014*), support the involvement of CXCR4 and its G-protein dependent signaling in CXCL12-induced Rac1 activation in cortical neurons.

## CXCL12 phosphorylates downstream mediators of Rac1 and results in a shift in the F/G actin ratio

Activation of Rac1 facilitates dendritic spine stability and actin polymerization through phosphorylation of downstream proteins, including PAK1, LIMK1, and cofilin. In CXCL12-treated neurons, we observed a time-dependent increase in phosphorylation in PAK1 (Thr423), LIMK1 (Tyr507/508), and cofilin (Ser3) (*Figure 5A*). In line with our observations regarding Rac1 activation, inhibition of CXCR4 and G$\alpha$i signaling completely blocked CXCL12-induced phosphorylation of PAK1, LIMK1, and cofilin (*Figure 5B and C*). Thus, CXCL12-mediated activation of Rac1 results in the phosphorylation of proteins critical for dendritic spine stability through CXCR4 and PTX-sensitive G$\alpha$i proteins.

The protein phosphatase Slingshot homolog 1 (SSH1) is known to regulate the phosphorylation status of both LIMK1 and cofilin (*Sparrow et al., 2012*). Dephosphorylated/active SSH1 exhibits dual activity in that it dephosphorylates and activates cofilin and, in parallel, dephosphorylates and inactivates LIMK1, resulting in the release of the brake imposed by LIMK1 on cofilin (*Niwa et al., 2002*). However, phosphorylation of SSH1 at Ser978 causes inactivation such that SSH1 can no longer dephosphorylate and inactivate LIMK1/activate cofilin (*Mizuno, 2013*). CXCL12 increased phosphorylation of SSH1 after one hour, demonstrating that not only does CXCL12 regulate the phosphorylation status of LIMK1 and cofilin through activation of Rac1 and PAK1, but also through inactivation of SSH1 (*Figure 5D*).

Increases in phosphorylated cofilin (Ser3) by either LIMK1 activation or SSH1 inactivation culminate in F-actin polymerization and a reduction in actin filament turnover. Therefore, we would expect a shift in the F/G-actin ratio in favor of F-actin in response to activation of this pathway by CXCL12.

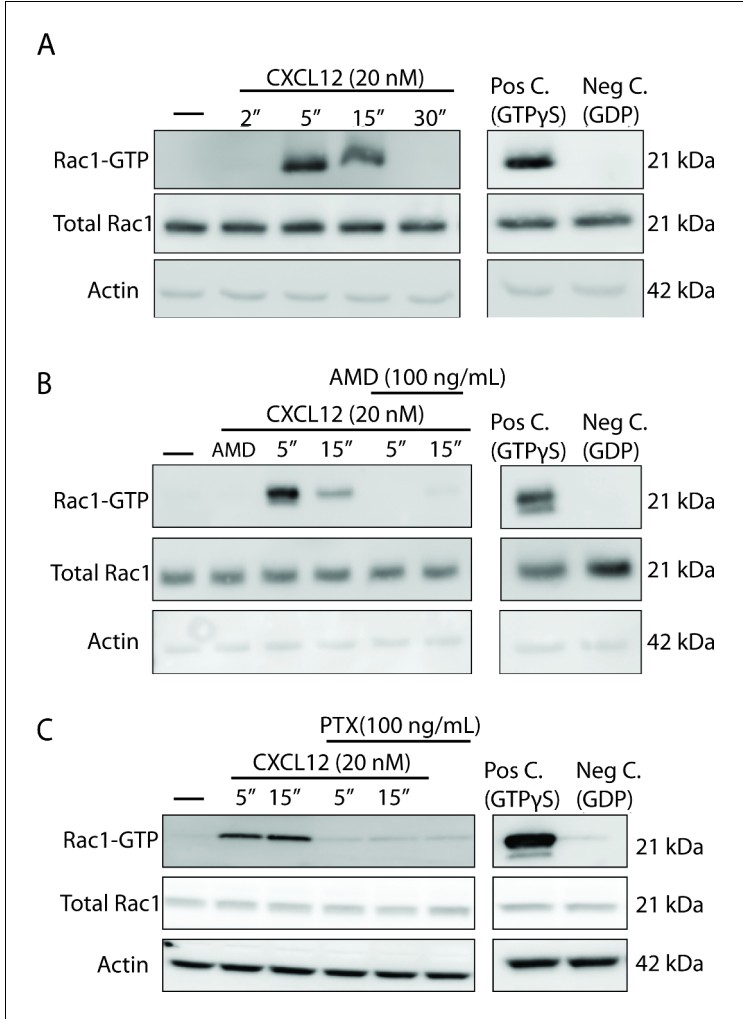

**Figure 4.** CXCL12 activates the small GTPase Rac1 in cultured cortical neurons in a CXCR4- and Gαi-dependent manner. (**A**) Cultured cortical neurons (21 DIV) were treated with CXCL12 (20 nM) for the indicated time points and subjected to pulldown using PAK agarose beads. Immunoblotting revealed a significant activation of Rac1 at 5 and 15 min and a return to baseline at 30 min. Positive and negative controls were performed by incubating lysates with GTPγS and GDP, respectively. N = 3 (**B**) Pretreatment with the CXCR4 antagonist, AMD3100 (100 ng/mL; 20 min) attenuated CXCL12-induced activation of Rac1. Positive and negative controls were performed by incubating lysates with GTPγS and GDP, respectively. N = 3 (**C**) Pretreatment with the Gαi inhibitor pertussis toxin (PTX, 100 ng/mL; 18 hr) completely prevented activation of Rac1 by CXCL12. Positive and negative controls were performed by incubating lysates with GTPγS and GDP, respectively. N = 3.

Cortical neurons treated with CXCL12 for one or two hours exhibited a four-fold increase in the F/G-actin ratio relative to vehicle-treated controls, indicative of a shift towards increased actin polymerization (*Figure 5E*). This functional change in the actin ratio is consistent with the time-dependent increase in dendritic spines induced by CXCL12 (*Pitcher et al., 2014*). These findings suggest that CXCL12-induced activation of the Rac1/PAK pathway in cortical neurons through CXCR4-, Gαi-dependent mechanisms, results in measurable, functional changes in actin polymerization, consistent with those needed for dendritic spine stability.

## CXCL12 specifically increases thin spine density via activation of Rac1

We observed activation of Rac1 and its downstream mediators by CXCL12 in cortical neurons, resulting in functional alterations in the F/G-actin ratio. Since these changes are associated with dendritic spine stability, we sought to investigate whether CXCL12 could upregulate a specific spine type. Thin spines appear to be likely targets for this signaling cascade due to the rapid F-actin

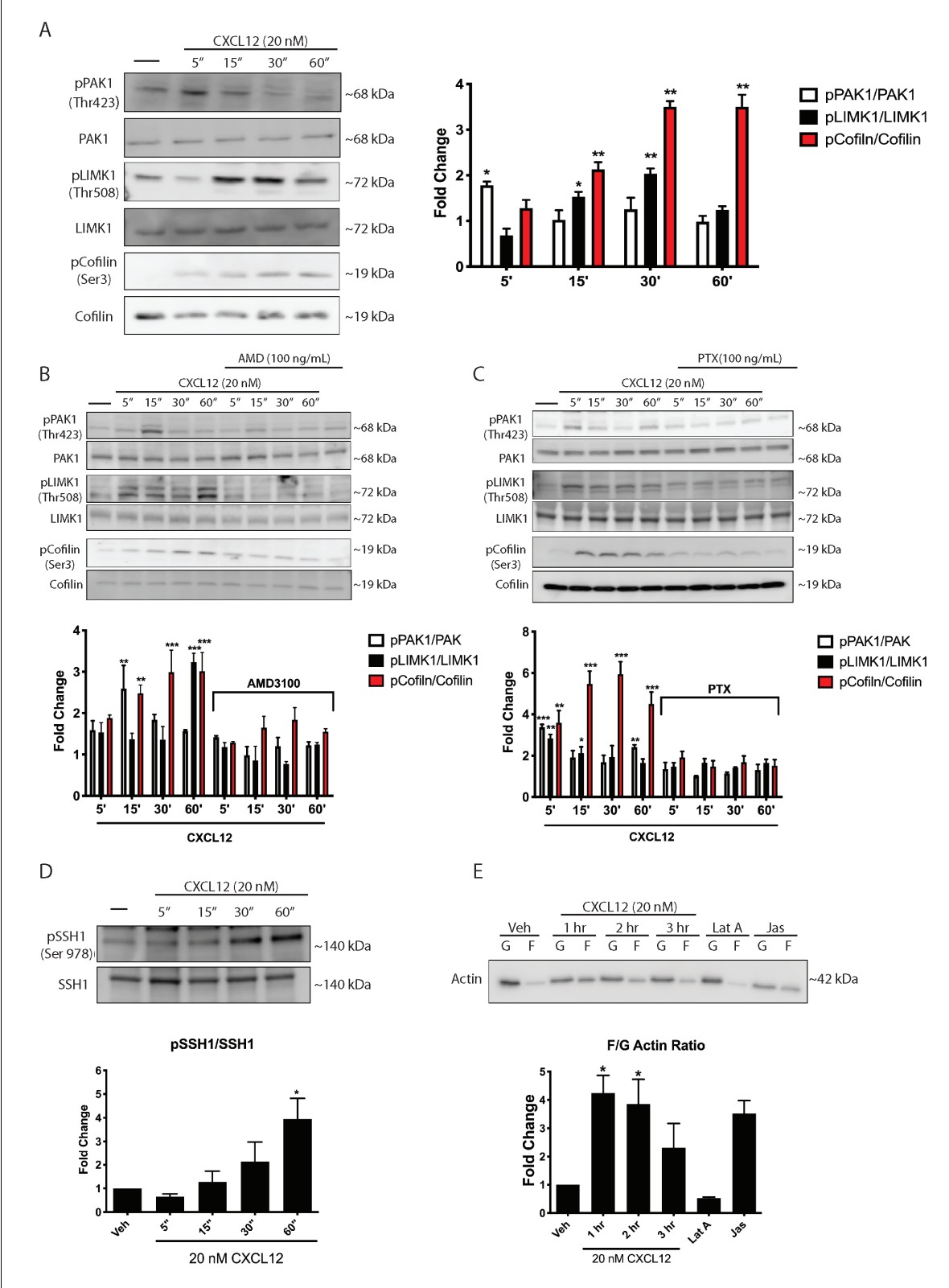

**Figure 5.** CXCL12 phosphorylates downstream mediators of the Rac1 pathway and results in changes to actin polymerization in cortical neurons. (A) Cultured cortical neurons (21 DIV) were exposed to CXCL12 (20 nM) for the indicated time points and the phosphorylation status of PAK1, LIMK1, and cofilin was examined. CXCL12 treatment resulted in a time-dependent increase in the phosphorylation of all three proteins. N = 3, *p<0.05, **p<0.01. (B) Pretreatment with AMD3100 (100 ng/mL; 20 min) blocked CXCL12-induced phosphorylation of Rac1 downstream mediators. N = 3, **p<0.01,

*Figure 5 continued on next page*

*Figure 5 continued*

***p<0.001. (C) Inhibition of Gαi signaling (via PTX) blocked the ability of CXCL12 to phosphorylate Rac1 downstream mediators. N = 3, *p<0.05, **p<0.01, ***p<0.001. (D) Following treatment with CXCL12, the protein phosphatase SSH1 is phosphorylated (and inactivated) in a time-dependent manner. N = 3, *p<0.05. (E) Separation of F and G-actin in cortical neurons revealed a shift in favor of F-actin following CXCL12 treatment. Latrunculin A (5 μM, 2 hr), a potent actin polymerization inhibitor, and jaspakinolide (5 μM, 2 hr), an inducer of actin polymerization, were used as internal controls for the assay. N = 3; *p<0.05.

The online version of this article includes the following source data for figure 5:

**Source data 1.** Densitometry statistical analysis.

polymerization and depolymerization that leads to their quick formation and elimination, often over the course of minutes. Rat primary cortical neurons (21 DIV) were exposed to CXCL12 (20 nM) for three hours; this treatment has previously been shown to elicit a peak effect of CXCL12 in cortical neurons (*Pitcher et al., 2014*). Consistent with our signaling data, CXCL12 specifically increased thin spine density on these neurons (*Figure 6A*). On the same segment of dendrite analyzed for thin spine density, we observed a subsequent reduction in stubby spine density. This suggests, together with our data regarding the shift in the F/G-actin ration, that spine stabilization is increased following CXCL12 treatment.

To determine whether CXCL12 depends on the activation of Rac1 to modulate dendritic spine density and morphology we used both pharmacologic and molecular approaches. First, we determined that pretreatment of neuronal cultures with the specific Rac1 inhibitor NSC23766 (*Gao et al., 2004*) (100 μg/mL; 15 mins) completely blocked CXCL12-induced activation of Rac1 (*Figure 6B*) and phosphorylation of its downstream mediators (*Figure 6C*). Next, we assessed alterations in dendritic spines by CXCL12 following either NSC23766 pretreatment or knockdown of Rac1 via shRNA (neurons infected at DIV 18 and treated on DIV 21). As shown in *Figure 6D*, the ability of CXCL12 to affect overall spine density and thin spine density was inhibited by blockade of Rac1 activation; furthermore, NSC23766 did not exhibit any effects on spine density or morphology on its own, demonstrating that acute inhibition of Rac1 does not negatively affect dendritic spines. Similarly, CXCL12 was unable to modulate spine density or morphology in Rac1 shRNA neurons (*Figure 6E*); in line with other studies (*Tashiro and Yuste, 2004*), prolonged inhibition of Rac1 expression by shRNA reduced total spine density (in either vehicle- and CXCL12-treated neurons). Taken together, these results point to the essential role of Rac1 activation in CXCL12-mediated dendritic spine alterations and provide the first mechanistic insight into this novel function of CXCL12.

## In vivo inhibition of Rac1 stimulation prevents CXCL12-induced alterations of dendritic spines and cognitive flexibility

Our in vitro studies elucidated the critical role of Rac1 activation in regulating CXCL12-mediated dendritic spine changes. We also demonstrated that CXCL12 reverts dendritic spines and cognitive flexibility deficits in vivo. Hence, activation of Rac1 and its downstream mediators are likely implicated in these restorative effects. The next set of studies aims to test this hypothesis. Immunohistochemical analysis of pPAK1 in NeuN+ cells in layer II/III of the mPFC was conducted and multispectral imaging was used to quantify changes in phosphorylation levels in individual neurons (~150–200 neurons per animal). While basal level of pPAK in HIV-Tg neurons was lower than WT, CXCL12 treatment in either genotype resulted in a robust increase in pPAK1 levels compared to vehicle-treated counterparts (*Figure 7A*). In Tg animals, the effect of CXCL12 on PAK1 activation was as evident as that seen in WT animals, resulting in a return to baseline phosphorylation levels. Since activation of PAK1 is intimately associated with alterations in dendritic spines and cognitive performance, we investigated the relationship between the two. This analysis showed that levels of pPAK1 in cortical neurons were positively correlated with overall dendritic spine density across all groups (*Figure 7B*), while it was negatively associated with trials to criterion on the shift to cue phase (*Figure 7C*). As predicted by the robust stimulation of pPAK1 in HIV-Tg animals, we observed no significant changes in total levels of Rac1 protein from frontal cortex lysates of WT and Tg animals (*Figure 7—figure supplement 1*). Importantly, in line with our observation that Tg animals have lower pPAK1 stimulation at baseline, pulldown studies from PFC tissue of untreated rats demonstrated reduced levels of active Rac1 in Tg rats compared to WT animals (*Figure 7—figure supplement 1*).

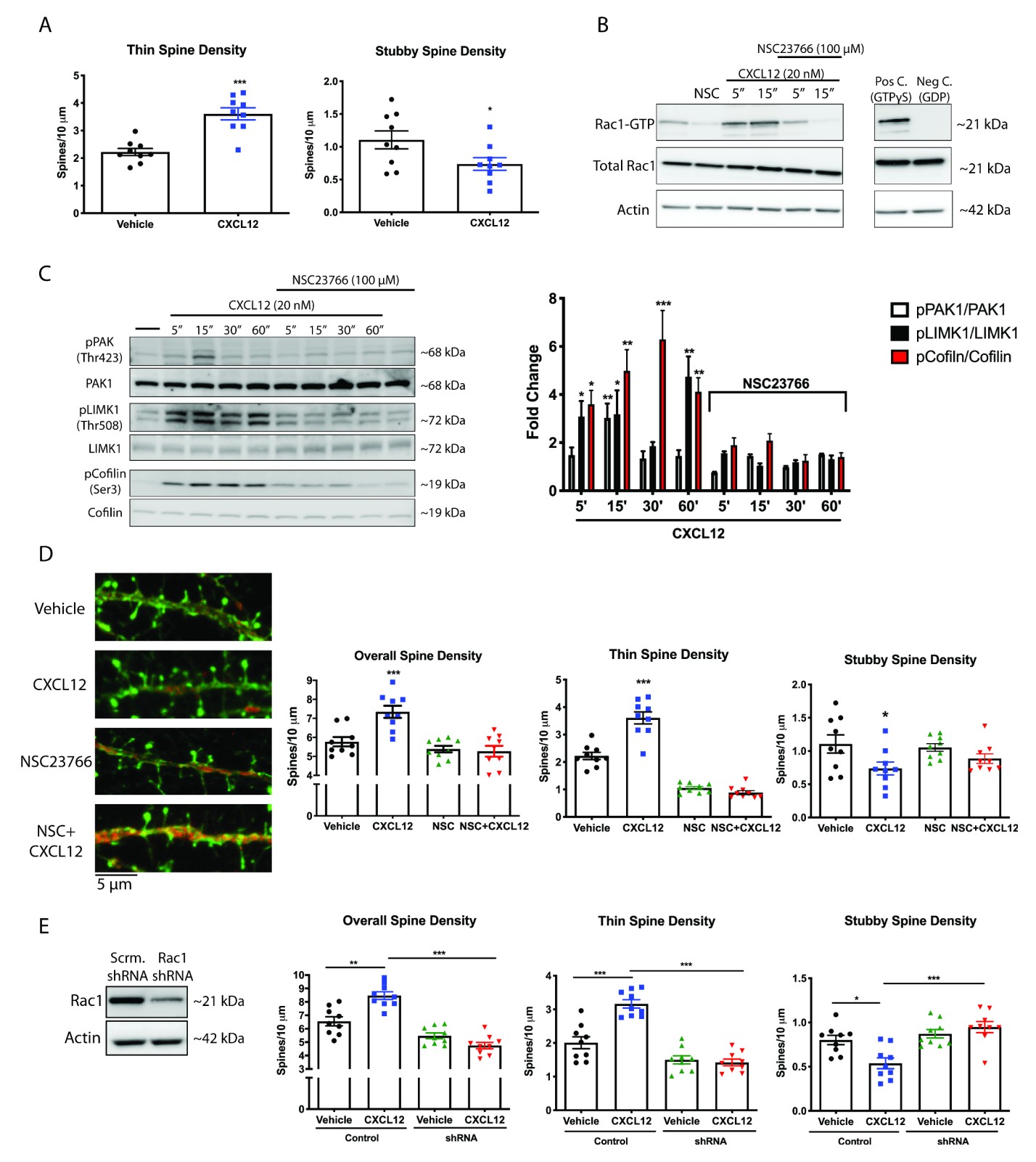

**Figure 6.** CXCL12 specifically modulates thin spine density via activation of Rac1.  (A) Cultured cortical neurons (21 DIV) were treated with CXCL12 (20 nM, 3 hr), resulting in a specific increase in thin spine density and a decrease in stubby spine numbers. N = 9 coverslips/group, 4 dendrites measured/coverslip and averaged into single data point, 3 separate experiments, *p<0.05, ***p<0.001. (B) Pretreatment with the specific Rac1 inhibitor NSC23766 (100 μM, 15 min) completely blocked CXCL12-induced activation of Rac1 in cortical neurons. N = 3 (C) Subsequently, inhibition of Rac1 activation by

*Figure 6 continued on next page*

*Figure 6 continued*

NSC23766 prevented phosphorylation of downstream mediators by CXCL12. N = 3, *p<0.05, **p<0.01, ***p<0.001. (D) Inhibition of Rac1 activation by NSC23766 blocked the ability of CXCL12 to modulate overall dendritic spine density, as well as thin and stubby spine density. N = 9 coverslips/group, 4 dendrites measured/coverslip and averaged into single data point, 3 separate experiments, *p<0.05, ***p<0.001. (E) Cortical neurons (18 DIV) were infected with control or Rac1-shRNA viral particles and GFP-positive neurons were analyzed (21 DIV) following treatment with either vehicle or CXCL12. Knockdown of Rac1 inhibited CXCL12-mediated alterations in spine density and morphology. N = 9 coverslips/group, 4 dendrites measured/coverslip and averaged into single data point, 3 separate experiments, *p<0.05, **p<0.01, ***p<0.001.

The online version of this article includes the following source data for figure 6:

**Source data 1.** In vitro raw data and statistical analysis.

We further investigated the role of Rac1 activation in regulating CXCL12-mediated dendritic spine dynamics and cognition in vivo. Prior to behavioral studies, the efficiency of in vivo Rac1

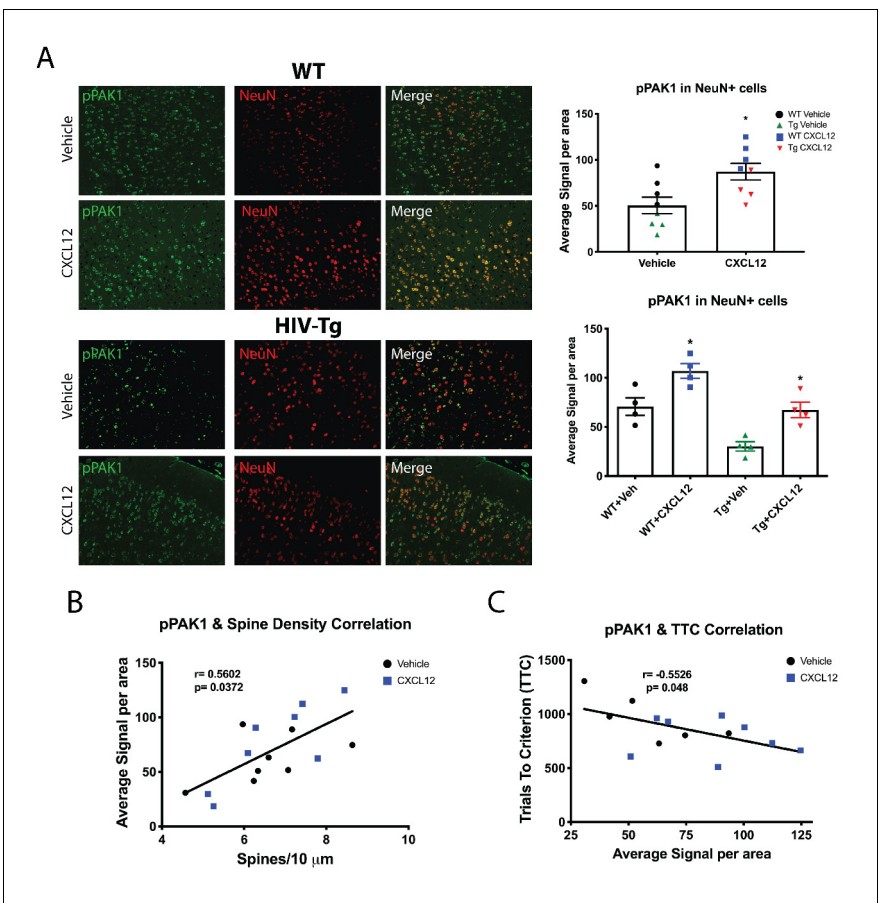

**Figure 7.** CXCL12 administration increases phosphorylation of PAK1 in layer II/III pyramidal neurons in the mPFC of male WT and HIV-Tg rats. (**A**) Sections from the contralateral hemisphere used for dendritic spine analysis were subjected to immunohistochemical and multispectral analysis. Both adult male WT and HIV-Tg rats treated with CXCL12 had a significant increase in pPAK1 levels in NeuN+ cells in the mPFC. N = 4/group, *p<0.05 for WT+Veh vs WT+CXCL12, *p<0.05 for WT+CXCL12 vs Tg+CXCL12, and *p<0.05 for Tg+Veh vs Tg+CXCL12. (**B**) Phosphorylation of PAK1 in the mPFC was positively associated with overall dendritic spine density in the same set of animals. N = 14 animals, Pearson's r = 0.5602, p=0.0372. (**C**) Levels of phosphorylated PAK1 are negatively correlated with the number of trials to criterion on the set shift phase. N = 14 animals, Pearson's r = −0.5526, p=0.048.

The online version of this article includes the following source data and figure supplement(s) for figure 7:

**Source data 1.** Tyramide IHC raw data and statistical analysis.

**Figure supplement 1.** HIV-Tg rats show reduced levels of activated Rac1 and no changes in total Rac1 protein expression in frontal cortex lysates.

inhibition was assessed. Four-month old male rats were implanted with a unilateral cannula as previously described and following a seven-day recovery period, once daily infusions of either vehicle (diH$_2$O) or NSC23766 (1 μg/μL or 2 μg/μL; 5 μL total volume) were initiated for eight days. At the end of the eight-day period, animals were sacrificed and one hemisphere was processed for Rac1 pulldown, while the other was utilized for immunohistochemical analysis of pPAK1. Immunoblotting revealed that both doses of NSC23766 significantly attenuated activation of Rac1 in frontal cortex lysates in a dose-dependent manner (*Figure 8—figure supplement 1*). Additionally, there was a dose-dependent decrease in pPAK1 in NeuN+ cells in the contralateral hemisphere (*Figure 8—figure supplement 1*). Thus, prolonged treatment with NSC23766 via ICV administration was well tolerated, and successfully downregulated activation of Rac1 and subsequent phosphorylation of its downstream mediators.

Based on the information generated from our initial experiments with NSC23766, we initiated studies to determine whether alterations in dendritic spines and cognitive flexibility induced by CXCL12 required Rac1 activation. Experiments were conducted in a similar manner as previous in vivo studies, though infusions of either vehicle or NSC23766 (2 μg/μL; 5 μL total volume) began one day prior to treatment with vehicle or CXCL12 (5 ng/μL; 5 μL total volume; *Figure 8—figure supplement 2*). Inhibition of Rac1 was sufficient to prevent the improvement in cognitive flexibility by CXCL12 as measured by the number of animals reaching criterion (Veh+CXCL12 vs. NSC+CXCL12, p=0.0498) and rate to reach criterion (F$_{(3,32)}$=12.6622, p<0.001) (*Figure 8A and B*), while no changes were observed in the first two phases of the task (*Figure 8—figure supplement 3*). In the same set of animals used for behavioral analysis, cotreatment with NSC23766 blocked upregulation of overall dendritic spine and thin spine density (*Figure 8C and D*) and prevented decreases in the number of stubby spines (*Figure 8E*). As previously observed, overall dendritic spine density in the mPFC across all four groups was inversely correlated to cognitive performance in set-shifting (Pearson's r = −0.7862, p=0.0121, α = 0.05; *Figure 8F*); this correlation became stronger when only thin spine density was considered (Pearson's r = −0.8350, p=0.0058, α = 0.05; *Figure 8G*).

Based on the findings in WT rats, we extended our studies on Rac1 inhibition to HIV-Tg rats in order to determine the role of Rac1 activation on the ability of CXCL12 to positively modulate cognitive flexibility and dendritic spines in a neuroinflammatory environment. Co-treatment with NSC23766 completely abrogated the ability of CXCL12 to enhance attentional set-shifting in HIV-Tg rats, with fewer animals reaching criterion (55% for Veh+CXCL12 compared to 25% for NSC+CXCL12, *Figure 9A*) and taking significantly longer to reach criterion (F$_{(3,32)}$=7.503, p=0.006, *Figure 9B*). As previously observed, inhibition of Rac1 activation had no deleterious effects on position discrimination and position reversal (*Figure 9—figure supplement 1*). Additionally, NSC23766 blocked the ability of CXCL12 to recover overall dendritic spine density (*Figure 9C*) and thin spine density (*Figure 9D*) in HIV-Tg rats, as well as decrease stubby spine density (*Figure 9E*). Despite no observed significant correlation between overall dendritic spine density and TTC in the shift to cue phase (Pearson's r = −0.2256, p=0.4808, α=0.05 *Figure 9F*), there was a strong negative correlation for thin dendritic spine density and shift to cue TTC (Pearson's r = −0.7787, p=0.0029, α = 0.05; *Figure 9G*). All together, these data demonstrate that CXCL12 depends on the activation of Rac1 to drive improvements in cognitive flexibility and dendritic spine number in physiological and neuroinflammatory conditions.

## Discussion

The present study takes the critical steps in defining the molecular pathways mediating CXCL12 regulation of dendritic spines in the mature brain. These novel CXCL12-induced neuronal changes are instrumental for rescuing cognitive flexibility. Importantly, this work demonstrates recovery of executive function and dendritic spines when impairment is already present, opening up new therapeutic avenues for HAND and other neuroinflammatory conditions.

### Complete rescue of behavioral and structural deficits in HAND

Characterization of executive function deficits in HIV-Tg rats revealed selective alterations in set-shifting, but not initial rule discrimination or reversal learning. Thus, neuronal injury induced by viral neurotoxins or low-level inflammation that results in appreciable cognitive dysfunction may be restricted to particular brain regions. The mPFC, in particular the PrL region, is known to mediate

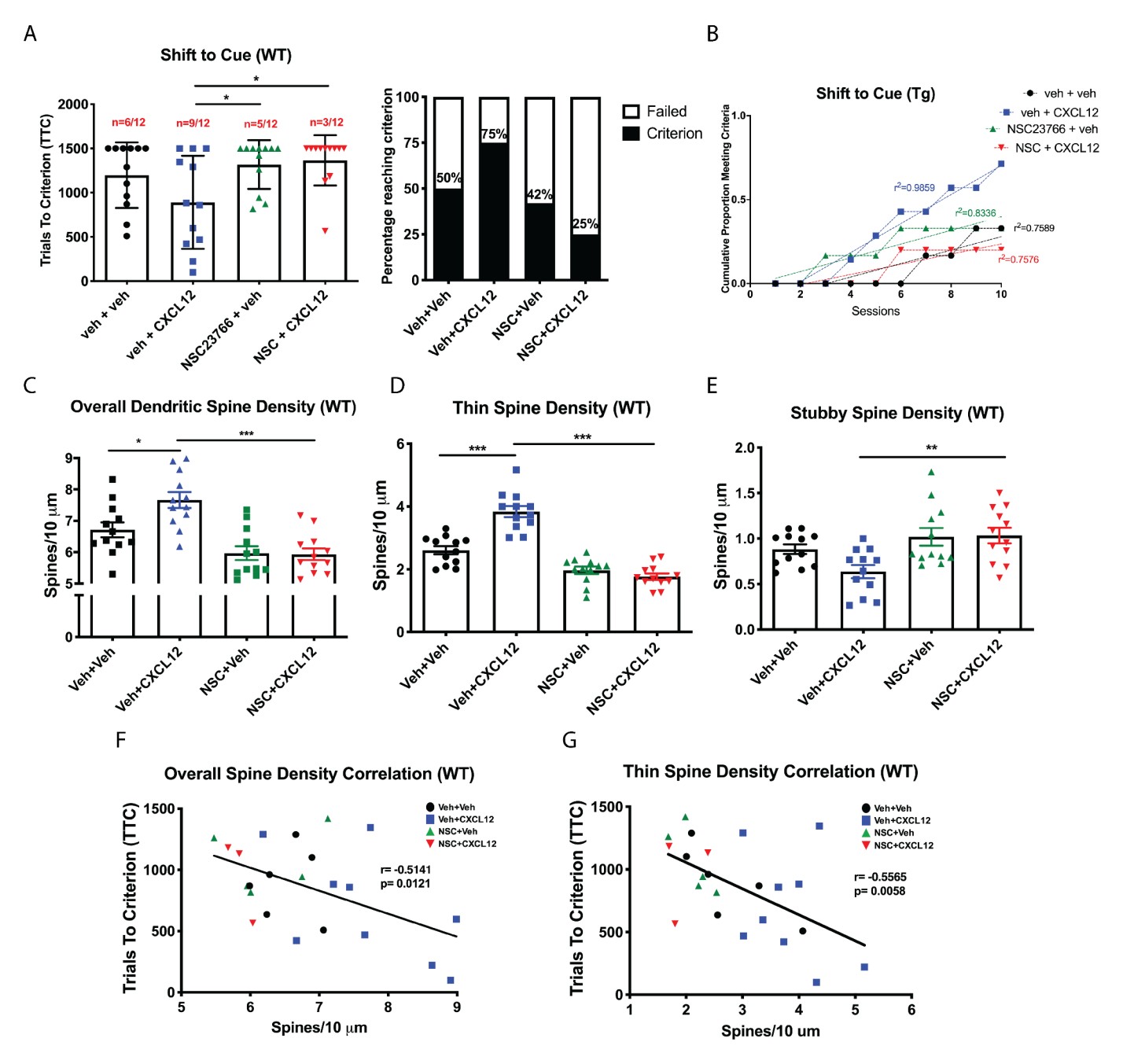

**Figure 8.** Blockade of Rac1 activation prevents alterations in cognitive flexibility and dendritic spines by CXCL12. (**A**) Treatment with NSC23766 mitigated the ability of CXCL12 to increase the number of animals reaching criterion on the shift to cue phase. N = 12 animals/group, p=0.0498. (**B**) The rate at which animals reached criterion was significantly delayed with NSC23766 pretreatment as assessed via linear regression. N = 12 animals/group, $F_{(3,32)=}12.6622$, p<0.001. (**C**) Blockade of Rac1 activation by NSC23766 prevents CXCL12-mediated upregulation of overall dendritic spine density in the mPFC. N = 12 animals/group, 8 dendrites measured for each animal and averaged into single data point, **p<0.01. (**D**) Additionally, NSC23766 prevented changes in thin spine density induced by CXCL12. N = 12 animals/group, 8 dendrites measured for each animal and averaged into single data point, ***p<0.001. (**E**) The ability of CXCL12 to decrease stubby spine density was blocked by NSC23766. N = 12 animals/group, 8 dendrites measured for each animal and averaged into single data point, **p<0.01. (**F**) As previously observed, overall dendritic spine density was negatively associated with trials to criterion on the set-shifting phase of the behavioral task. N = 23 animals, Pearson's r = −0.7862, p=0.0127. (**G**) This relationship became even stronger when only thin spine density was considered. N = 23 animals, Pearson's r = −0.8350, p=0.0051.

The online version of this article includes the following source data and figure supplement(s) for figure 8:

**Source data 1.** WT ±CXCL12/NSC23766 raw data and statistical analysis.

*Figure 8 continued on next page*

*Figure 8 continued*

**Figure supplement 1.** NSC23766 inhibits activation of Rac1 in vivo.

**Figure supplement 2.** Experimental timeline for behavior and ICV infusions in *Figures 8* and *9*.

**Figure supplement 3.** NSC23766 treatment did not alter performance on position discrimination or position reversal in WT rats.

strategy shifting, as inactivation or lesions to this area specifically impairs this behavioral phenotype while sparing both initial learning and reversal learning (*Birrell and Brown, 2000*; *Floresco et al., 2008*). On the other hand, damage to the anterior cingulate cortex or orbitofrontal cortex results in deficits in initial rule acquisition or reversal learning respectively (*Hamilton and Brigman, 2015*). Based on our behavioral data, we predict minimal changes in dendritic spines in these regions and ongoing studies in the same animal model are exploring whether other brain regions besides the mPFC display deficits in spine number and morphology. Regional analysis revealed selective alterations in the motor cortex, while there are marginal changes in the neighboring somatosensory cortex (*Nash et al., 2019*). In line with clinical imaging studies (*Sanford et al., 2018*; *Sanford et al., 2017*), dendritic damage in the context of HIV infection seems to be region specific and very subtle. Importantly, our data are in line with others who have demonstrated subtle structural deficits of dopaminergic medium spiny neurons in the nucleus accumbens of female HIV-Tg rats (*Roscoe et al., 2014*).

Using DiOlistic staining coupled with Neurolucida 360 software, we were able to successfully classify individual spines into one of the four main categories based on established parameters (*Rodriguez et al., 2008*). This revealed a significant increase in stubby spine density, associated with decreases in both thin and mushroom spine density, on layer II/III pyramidal neurons in the mPFC in HIV-Tg rats. This demonstrates an overall shift in spine morphology, whereby there is a greater proportion of immature (e.g. stubby) spines remaining on these neurons and fewer mature (e.g. thin and mushroom) ones. Combined with the overall reduction in spine density, this finding has major implications for neurotransmission and cognitive function. Stubby spines are considered immature structures due to their usual prevalence during early postnatal development and scarcity in the adult CNS (*Harris, 1999*). Their lack of a distinctive head and neck suggests that these spines are unable to properly sequester incoming electrical and chemical signals, leading to inappropriate propagation of downstream pathways. Particularly, a dendritic spine's head and neck configuration is essential to properly compartmentalize incoming $Ca^{2+}$ influx through glutamate receptors in order to prevent excitotoxicity from occuring (*Dong et al., 2009*). Excitotoxic neuronal injury is a common endpoint for viral proteins and inflammatory mediators (*Saylor et al., 2016*) and the higher proportion of stubby dendritic spines may represent a common modality by which these proteins can induce damage. Additionally, electrophysiological data from HIV-Tg rats demonstrated that layer V pyramidal neurons from the PrL region of the mPFC are hyper-excitable compared to neurons from non-Tg controls, as indicated by reduced rheobase, spike amplitude, an inwardly-rectifying $K^+$ influx, increased number of action potentials, and a trend towards aberrant firing (*Khodr et al., 2016*; *Wayman et al., 2016*). While our studies focused on dendritic spine alterations in layer II/III neurons, additional data from the same set of animals demonstrated similar deficits in layer V cells and recovery by CXCL12 administration (data not shown) suggesting that injury in the mPFC is not layer-specific. Thus, the increase in stubby spine density on these neurons may result in aberrant synaptic transmission and hyper-excitability, leading to inappropriate activation of $Ca^{2+}$-dependent enzymes and structural damage to the neuron. This would result in considerable circuit alterations and behavioral dysfunction.

Notably, treatment with exogenous CXCL12 completely restored the reduction in dendritic spine density observed in the mPFC of HIV-Tg rats, as well as the associated dysfunction in set-shifting. To our knowledge, this is the first study reporting recovery of both structural and behavioral deficits in the HIV-Tg rat model. Importantly, studies were conducted at an age where animals were already impaired, demonstrating that dendritic spine injury and cognitive dysfunction are completely reversible. This has important implications for HIV+ patients on ART where viral replication has been suppressed, but viral proteins and inflammatory mediators may still be produced, especially in the CNS, and continue to induce damage (*Saylor et al., 2016*). It also supports the notion that targeting homeostatic mediators of neuronal function and synaptic transmission that are disrupted during HIV

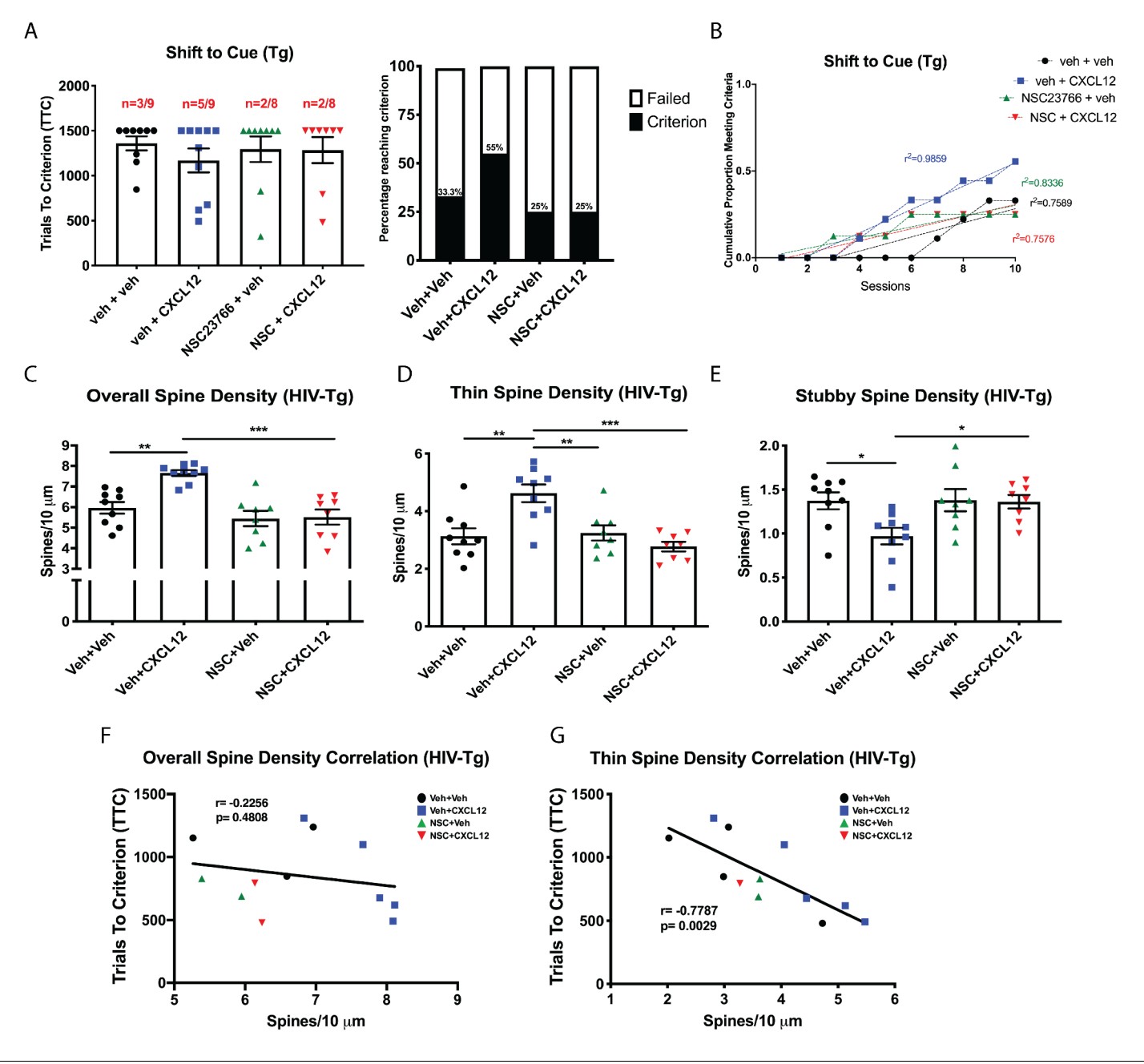

**Figure 9.** CXCL12 depends on Rac1 activation to rescue cognitive flexibility and dendritic spine density in the HIV-Tg rat. (A) Inhibition of Rac1 activity blocked the ability of CXCL12 to positively enhance cognitive flexibility in an attentional set-shifting task. N = 9 for veh+veh, N = 9 for veh+CXCL12, N = 8 for NSC+veh, and N = 8 for NSC+CXCL12. (B) NSC23766 co-treatment significantly attenuated the rate at which animals reached criterion on the shift to cue phase as assessed by linear regression. N = 9 for veh+veh, N = 9 for veh+CXCL12, N = 8 for NSC+veh, and N = 8 for NSC+CXCL12, $F_{(3,32)}$=7.503, p=0.0006. (C) Blockade of Rac1 activation abrogated the effect of CXCL12 on overall dendritic spine density in HIV-Tg rats. N = 9 for veh +veh, N = 9 for veh+CXCL12, N = 8 for NSC+veh, and N = 8 for NSC+CXCL12, eight dendrites measured for each animal and averaged into single data point, **p<0.01, ***p<0.001. (D) CXCL12's impact on thin spines was mitigated by co-treatment with NSC23766. N = 9 for veh+veh, N = 9 for veh +CXCL12, N = 8 for NSC+veh, and N = 8 for NSC+CXCL12, eight dendrites measured for each animal and averaged into single data point, **p<0.01, ***p<0.001. (E) NSC23766 prevented decreases in stubby spine density mediated by CXCL12 treatment. N = 9 for veh+veh, N = 9 for veh+CXCL12, N = 8 for NSC+veh, and N = 8 for NSC+CXCL12, eight dendrites measured for each animal and averaged into single data point, *p<0.05. (F) There was no significant relationship observed between overall dendritic spine density and trials to criterion on the set-shifting phase of the behavioral task. N = 12 animals, Pearson's r = −0.2256, p=0.4808. (G) Thin spine density was negatively associated with the number of trials to criterion in the shift to cue phase of the behavioral task. N = 12 animals, Pearson's r = −0.7787, p=0.0029.
*Figure 9 continued on next page*

*Figure 9 continued*

The online version of this article includes the following source data and figure supplement(s) for figure 9:

**Source data 1.** HIV-Tg ±CXCL12/NSC23766 raw data and statistical analysis.

**Figure supplement 1.** Inhibition of Rac1 activation by NSC23766 does not alter performance on position discrimination or position reversal in HIV-Tg rats.

infection may be a valid therapeutic strategy for HAND patients. Our data also suggest that restoring highly plastic thin spines is a key driver in regulating cognitive function mediated by the mPFC. This is the first time this phenomenon has been reported in a rodent model of HAND, suggesting that therapeutics to target this spine type are an appropriate strategy to combat the deficits seen in these patients. Interestingly, the extended treatment with CXCL12 did not reveal any side effects in both WT and Tg animals, indicating that potential effects of the chemokine on other cellular targets (*Guyon, 2014*) do not interfere with its neuroprotective function - at least within the time frame of this study. Most importantly, our work dissecting out the molecular mechanisms of CXCL12-mediated spine alterations points to the key involvement of the Rac1/PAK pathway. Therefore, therapies targeting this pathway specifically in neurons would be beneficial in HAND and other types of cognitive dysfunction, such as aging (*Hao et al., 2006*).

## Role of Rac1/PAK pathway in CXCL12-enhanced cognitive performance

Though the chemokine CXCL12 is well known for its homeostatic action in the developing brain (*Lysko et al., 2014*), its relevance in the mature brain has been mainly associated with inflammatory responses. Here, we demonstrated that CXCL12 activates the small GTPase Rac1 and its downstream mediators to modulate dendritic spine density in mature, excitatory cortical neurons. Activation of this pathway is associated with actin polymerization and this resulted in a specific increase in thin spine density on cortical neurons. The involvement of the Rac1/PAK pathway in mediating the effects of CXCL12 on actin polymerization has not been described in cortical neurons. Our data show that Rac1 activation is the exclusive mediator of CXCL12's alterations on spine density and morphology, suggesting that targeted therapeutics to selectively activate this pathway through CXCR4 could recover dendritic spine loss. CXCL12 not only activated the Rac1/PAK pathway but also modulated the phosphorylation status of the phosphatase Slingshot homolog 1. This dual regulation of cofilin demonstrates that CXCL12 acts on multiple components of this pathway to exert control on actin polymerization. This also provides additional targets to positively modify dendritic spines. While it is presently unclear whether CXCL12 regulates phosphorylation of SSH1 in cortical neurons in a direct manner, there is evidence in non-neuronal cells that the chemokine can inhibit dephosphorylation of SSH1 via the 14-3-3 protein family (*Chen et al., 2015*; *Chernock et al., 2001*). 14-3-3 proteins are a family of highly conserved proteins that are abundantly expressed in the adult brain (*Dougherty and Morrison, 2004*). They inhibit SSH1 phosphatase activity by binding to phosphorylated serine residues and sequestering SSH1 in the cytoplasm (*Nagata-Ohashi et al., 2004*; *Soosairajah et al., 2005*). Since our current data regarding CXCL12-induced activation of the Rac1/PAK pathway are in line with studies in non-neuronal cells, the mechanisms leading to SSH1 phosphorylation might also be conserved. However, this needs to be investigated.

Inactivation of Rac1 via a small molecule inhibitor completely blocked the beneficial effects of CXCL12 on set shifting as measured by trials to criterion and the rate to reach criterion. Furthermore, changes in spine density and morphology induced by daily CXCL12 treatment were completely dependent on its ability to activate Rac1 in the mPFC. Interestingly, blockade of Rac1 activation had no perceived effects on the first two phases of the behavioral task, suggesting specificity of the small GTPase in functions mediated by the mPFC but this warrants further investigation. Nevertheless, to our knowledge, this is the first study to connect Rac1 activation with behavioral flexibility, further underscoring the critical role of the small GTPase in regulating synaptic function. Additionally, our data support the notion that utilizing therapeutics to modulate Rac1/PAK signaling may be beneficial in reversing structural and behavioral deficits in the mPFC.

## CXCL12 in the mature CNS

The role of CXCL12/CXCR4 in the developing CNS is well established. On the other hand, the effects of CXCL12 on mature neurons, specifically in the context of synaptic connectivity, are not completely understood. In cortical neurons, induction of Rac1 is associated with F-actin polymerization and spine stability (*Lin and Koleske, 2010*). This is in agreement with our data demonstrating a shift in F/G-actin ratio following CXCL12 treatment, as well as the specific increase in transient thin spines. Upregulation of thin spines, which at least initially only contain NMDARs and are considered 'silent synapses', would not be expected to result in significant alterations in synaptic transmission. However, as previously discussed, thin spines control neuronal signaling via regulation of calcium-dependent processes. Moreover, our previous work demonstrated that CXCL12 altered both sEPSC and mEPSC amplitude in whole-cell patch clamp slice recordings (*Pitcher et al., 2014*). Changes in mEPSCs amplitude typically reflect alterations to postsynaptic receptors, suggesting that CXCL12 is contributing to the maturation of dendritic spines. Furthermore, Rac1 activation is implicated in spine head morphing, or twitching, that promotes the maturation of synaptic contacts (*Holtmaat and Svoboda, 2009*; *Tashiro and Yuste, 2004*). While we did not observe any changes in mushroom spine density by CXCL12 in vitro or in vivo, this is not surprising given that mushroom spines are the most mature and stable spines, typically implicated in long-term memory and potentiation (*Bourne and Harris, 2007*). Ongoing studies in the laboratory are investigating whether CXCL12 alters levels of post-synaptic density protein 95 (PSD-95), a necessary scaffolding protein present in mature synapses. Shifts in the actin ratio towards F-actin results in an increase in spine motility and spine head enlargement; thus, this chemokine plays a critical and unique role in regulating synaptic connectivity via modulation of the number of contacts as well as their maturation.

It is important to note that the relationship between CXCL12 and CXCR4 is not exclusive, as CXCL12 can also interact with another receptor, atypical chemokine receptor 3 (ACKR3), also known as CXCR7. ACKR3 contains a mutated 'DRYLAIV' motif (*Sierro et al., 2007*; *Thelen and Thelen, 2008*), a region critical for G-protein coupling, and thus does not demonstrate G-protein-dependent signaling. Early reports supported a scavenger role, acting to internalize and degrade CXCL12, thereby regulating extracellular concentration. Removal of ACKR3 causes a significant increase in extracellular levels of CXCL12, which favors its binding to CXCR4 and triggers CXCR4 endocytosis and degradation (*Sánchez-Alcañiz et al., 2011*). Furthermore, ACKR3 can form heterodimers with CXCR4 and regulate CXCR4-mediated signaling; however, the presence of these heterodimers occurring in vivo has been called into question (*Levoye et al., 2009*; *Luker et al., 2012*). While the above-mentioned evidence demonstrates that ACKR3 may modulate CXCR4 signaling pathways and responsiveness, the demonstrated absence of cell-surface ACKR3 in mature neurons (ACKR3 is only transiently expressed on the surface of developing neurons) (*Shimizu et al., 2011*), suggests a predominant role of CXCR4 in CXCL12-mediated effects on dendritic spines and cognition.

While CXCR4 expression has been detected on other cell types of the CNS, including astrocytes (*Bajetto et al., 1999*), microglia (*Lipfert et al., 2013*), and oligodendrocytes (*Maysami et al., 2006*), CXCL12's effects on neurons are observed both in the presence and absence of glia (*Pitcher et al., 2014*; *Sengupta et al., 2009*), and its ability to regulate dendritic spines in cultured neurons is replicated in the intact brain. Thus, the chemokine's interaction with neuronal CXCR4 is sufficient to induce these changes. Our current studies aim to establish whether CXCL12 is modulating specific neuronal subpopulations (i.e. excitatory or inhibitory neurons). CXCL12 is known to alter GABAergic neurotransmission (*Bhattacharyya et al., 2008*; *Guyon and Nahon, 2007*), which may result in dendritic spines changes on excitatory neurons (*Chiu et al., 2013*). Likewise, recent evidence has also shown that CXCL12 targets parvalbumin (PV)+ basket interneuron axons and inhibitory perisomatic synapses, which modulates neuronal excitability (*Wu et al., 2017*). Ongoing experiments are utilizing dCas9-base CRISPR interference (CRISPRi) to selectively knockdown CXCR4 in either excitatory (CamKIIα+) or inhibitory (mDlx+) neurons and determine the contribution of each neuronal subpopulation to dendritic spine dynamics.

In conclusion, these studies open the possibility of creating therapeutics to target synaptic and cognitive deficits in HAND patients via restoration of endogenous neuroprotective pathways. While drug discovery involving chemokine receptors has primarily focused on developing antagonists, our work suggests that CXCR4 agonists may be beneficial in alleviating structural and behavioral dysfunction. The in vitro studies reported here dissected out the molecular players regulating CXCL12-

induced spine changes, as well as identified the specific spine type targeted by the chemokine. The in vivo experiments validated our mechanistic studies and confirmed the involvement of Rac1 signaling in CXCL12-regulated dendritic spine and cognitive changes. Taken together, these data suggest that targeting these signaling pathways may represent a valid strategy for neurorestorative therapies for HAND and other diseases characterized by spine loss and/or CXCR4 dysregulation, such as Alzheimer's disease (AD) and schizophrenia. This represents a major paradigm shift, as previous attempts against HAND or AD focused on inhibition of excitotoxicity through targeting of glutamate receptors, transporters, or ion channels. Interestingly, small molecule CXCR4 agonists that may be exploited for further preclinical studies are now available (*Mishra et al., 2016*).

# Materials and methods

## Key resources table

| Reagent type (species) or resource | Designation | Source or reference | Identifiers | Additional information |
|---|---|---|---|---|
| Strain, strain background (*R. Norvegicus*) | F344/NHsd (Male) | Envigo and University of Maryland | RRID:RGD_61109 | |
| Strain, strain background (*R. Norvegicus*) | HSD:HIV-1 (F344) (Male) | Envigo and University of Maryland | | |
| Cell line (*H. sapiens*) | HEK293T | ATCC | RRID:CVCL_0063 | |
| Transfected construct (*R. Norvegicus*) | Rac1 shRNA | Origene Technologies | Cat # TL712781 | |
| Recombinant DNA reagent | pCMVR8.74 | AddGene | RRID: Addgene_22036 | |
| Recombinant DNA reagent | pMD2.G | AddGene | RRID: Addgene_12259 | |
| Antibody | Anti-pPAK1 rabbit polyclonal (Thr423) | Cell Signaling Technology | RRID: AB_330220 | 1:1000 |
| Antibody | Anti-PAK1 rabbit polyclonal | Cell Signaling Technology | RRID: AB_330222 | 1:1000 |
| Antibody | Anti-pLIMK1 rabbit polyclonal (Thr507/508) | EMD Millipore | RRID: AB_568901 | 1:1000 |
| Antibody | Anti-LIMK1 rabbit polyclonal | Cell Signaling Technology | RRID: AB_2281332 | 1:1000 |
| Antibody | Anti-pCofilin rabbit polyclonal (Ser3) | Cell Signaling Technology | RRID: AB_330238 | 1:1000 |
| Antibody | Anti-Cofilin rabbit monoclonal | Cell Signaling Technology | RRID: AB_10622000 | 1:1000 |
| Antibody | Anti-pSSH1L rabbit polyclonal (Ser978) | ECM Biosciences | RRID: AB_10553849 | 1:500 |
| Antibody | Anti-SSH1 rabbit monoclonal | Cell Signaling Technology | RRID: AB_2798263 | 1:1000 |
| Antibody | Anti-Rac1 | Cell Biolabs | Included in #STA-401–1 | 1:6000 |
| Antibody | Anti-β-actin rabbit polyclonal | Sigma-Aldrich | RRID: AB_476693 | 1:6000 |

*Continued on next page*

*Continued*

| Reagent type (species) or resource | Designation | Source or reference | Identifiers | Additional information |
|---|---|---|---|---|
| Antibody | Anti-MAP2 rabbit polyclonal | EMD Millipore | RRID: AB_91939 | 1:1000 |
| Antibody | Anti-pPAK1 rabbit polyclonal (Thr423) | Invitrogen | RRID: AB_2554427 | 1:25 |
| Antibody | Anti-NeuN rabbit monoclonal | Cell Signaling Technology | RRID: AB_2651140 | 1:400 |
| Antibody | Goat polyclonal anti-rabbit Alexa Fluor 568 | Invitrogen | RRID:AB_143157 | 1:250 |
| Peptide, recombinant protein | Rat CXCL12 | Peprotech | Cat# 400-32A | |
| Commercial assay or kit | Rac1 Activity Assay Kit | Cell Biolabs | Cat# STA-401–1 | |
| Chemical compound, drug | NSC23766 | Tocris | Cat# 2161 | |
| Software, algorithm | GraphPad Prism | GraphPad Software | RRID: SCR_015382 | Version 8.0 |
| Software, algorithm | Neurolucida 360 | MBF Biosciences | RRID: SCR_016788 | Version 2017.01.4 |
| Software, algorithm | Nuance Multispectral Imaging Systems | Nuance | RRID: SCR_015382 | Version 2.10 |
| Software, algorithm | Med-PC | Med Associates | RRID: SCR_012156 | Version 4.1 |
| Other | Helios Gene Gun | Bio-Rad | Cat# 1652411 | |
| Other | Phalloidin Alexa Fluor 488 | Invitrogen | Cat # A12379 | 1:400 |
| Other | Alexa Fluor conjugated tyramide 488 | Invitrogen | Cat # B40953 | |
| Other | Alexa Fluor conjugated tyramide 555 | Invitrogen | Cat# B40955 | |
| Other | DiI Stain | Invitrogen | Cat# D282 | |
| Other | Hoechst | Invitrogen | H3570 | 1 µg/mL |

## Animals

Adult male wild-type (WT; F344/NHsd; n = 79) and HIV-1 transgenic rats (Tg; HSD:HIV-1(F344); n = 59, approximately 4 months old) were utilized for behavioral and dendritic spine analysis. Animals were purchased from Envigo Laboratories (Indianapolis, IN) until the license for breeding HIV-Tg rats was returned to the University of Maryland, who originally developed this transgenic line (*Reid et al., 2001*). Rats were singly housed in isolation in our Association for Assessment and Accreditation of Laboratory Animal Care-accredited barrier facilities in accordance with the National Institutes of Health guidelines and institutional approval by the Institutional Animal Care and Use Committee. Animals were food-restricted one week prior to the initiation of lever pressing training.

## Cannula implantation and intracerebroventricular administration

Rats were implanted stereotaxically under isoflurane anesthesia and ketamine/xylazine (50 mg/kg; 10 mg/kg, PennVet), with 26-gauge stainless-steel guide cannulas (Plastics One) placed 0.96 mm posterior and 2.00 mm lateral to bregma, 3.5 mm below the surface of the cranium. Four stainless-steel screws (#0–80) were placed around the cannula and acrylic dental cement was used to anchor them. Following surgery, animals were allowed seven days to recover before the initiation of infusions and behavioral testing. Animals received once daily infusions of either vehicle (0.1% BSA in PBS, 5 µL total volume), CXCL12 (5 ng/µL in 0.1% BSA in PBS, 5 µL total volume), or NSC23766 (2 µg/µL in diH$_2$O, 5 µL total volume) via a micropump set for an infusion rate of 0.5 µL/min throughout the duration of behavioral testing.

## Attentional set-shifting task

Reversal learning and strategy shifting were assessed using an automated operant-based approach whereby the behavioral tasks occurred within operant chambers controlled by custom software programs (*Brady and Floresco, 2015*). Each chamber was equipped with a house light, fan, two retractable levers, a tone generator above each lever, a pellet dispenser, a stainless-steel pellet trough located between the levers, and a panel light just above the pellet trough. Food-restricted rats were initially trained to respond by lever pressing under a fixed-ratio (FR1) schedule of reward presentation (45 mg sucrose pellet, Bio-Serv, Flemington, NJ). Side biases were avoided by presenting each lever an equal number of times in a pseudorandom order. Following training, the 3-phase task was conducted, consisting of position discrimination, position reversal, and rule shifting. During position discrimination, one lever (left or right) was designated as the 'correct' choice. Both levers were presented but only responses on the 'correct' lever were rewarded while responses on the other had no programmed consequences. An inter-trial interval of 20 s followed each response, during which time the levers were retracted. Rats underwent position discrimination until they attained the criterion of 10 consecutive correct trials with a maximum of 150 trials per day. One day following completion of the discrimination phase, the rats performed a reversal learning task in which the opposite lever of the 'correct' choice from the discrimination phase was now rewarded. Twenty reminder trials of the previous rule were conducted prior to the reversal trials. As with position discrimination, rats underwent reversal learning until they attained 10 consecutive correct trials. After successful completion of the reversal task, the strategy shift task was initiated on the following day. In this phase both levers were presented in each trial. In addition, an audible tone was presented above one of the two levers. The 'correct' lever, which resulted in reward presentation, was that which was associated with the tone, regardless of position (left or right) within the chamber. Tone, rather than a cue light, was used in this phase to account for any inherent visual impairment due to cataract development in Tg rats. Rats conducted the strategy shift phase until they attained the criterion of 10 consecutive correct trials, with a maximum of 150 trials on a single day. If animals failed to reach criterion by day 10 of the shift to cue phase, the task was terminated. Cognitive performance was determined by measuring the number of trials needed to reach criterion during each phase of the task, as well as the rate at which they reached criterion.

## Neuronal cultures

Primary cortical neurons were obtained from E17 embryos from Holtzmann rats and cultured in Neurobasal medium as previously reported (*Brewer, 1997*; *Pitcher et al., 2014*; *Sengupta et al., 2009*). To further reduce the growth of non-neuronal cells, cytosine arabinoside (1 µM) was added to the cultures. This results in a virtually glia-free culture (>95% neurons) as assessed by expression of neuronal and non-neuronal markers. Briefly, neurons were plated at a density of $1 \times 10^6$ on 60 mm dishes or 35,000 on 15 mm coverslips in Neurobasal medium containing B27 (2%) and horse serum (2%). Medium was replaced after 3 hr of plating with serum-free Neurobasal medium supplemented with B27, Glutamax (0.5 mM), and glutamic acid (25 µM). On the fourth and eighth day in vitro (DIV), the culture medium was replaced again with Neurobasal medium supplemented with B27 and Glutamax. On the fourteenth day of culture, half of the media were changed with Neurobasal medium supplemented with B27 and Glutamax. All experiments were conducted at DIV 21.

## Infection of cortical neurons

Cortical neurons (DIV 18) were infected with either scramble shRNA or Rac1 shRNA (5'-CGAGGAC TCAAGACAGTGTTTGATGAAGC-3'; OriGene Technologies) lentiviral particles. Lentivirus was produced in HEK293T cells (cultured in DMEM and 10% fetal bovine serum) by transfecting cells with either the control or Rac1 shRNA plasmid vector (pGFP-C-shLenti), pCMVR8.74 (Addgene #22036), and pMD2.G (Addgene #12259) via calcium phosphate. Supernatants were collected on day 2 and 3 post-transfection and lentiviral particles were isolated via ultracentrifugation (2 hr at 50,000 g, 16°C) and resuspended in 1X PBS. Purified virus was added directly to cortical neurons and resulted in 30–40% infection efficiency. Levels of Rac1 protein were assessed via Western blot (as described below) after 3 days of infection.

## Western blot

For whole-cell lysates, cell lysates were prepared after drug treatment as previously described *Sengupta et al. (2009)*. Primary neurons were washed in ice-cold PBS, collected in lysis buffer (150 mM NaCl, 50 mM Tris, 0.5% Na deoxycholate, 0.1% SDS, 10 mM $Na_4P_2O_7$, 5 mM EDTA, 1% Triton-X 100, 1 mM DTT, and protease/phosphatase inhibitors), and then incubated for 30 min on a rotor at 4°C. After 30 min, lysates were spun at 20,800 $x$ $g$ for 10 min and the protein concentration of the supernatants was determined using bicinchoninic acid (BCA) assay following the instructions of the manufacturer (Pierce). Equal amounts of protein (30–40 µg/lane) were loaded for SDS-PAGE followed by immunoblotting.

For tissue homogenates, brain cortices were rapidly removed, and a 5 mm portion of the frontal cortex was separated using a brain matrix. Tissue was dissociated by pipetting and incubated for 1 hr at 4°C with frequent vortexing. Lysates were then centrifuged at 20,800 $x$ $g$ for 10 min at 4°C and protein concentration was assessed via BCA assay. Equal amounts of protein (30–40 µg/lane) were used for SDS-PAGE followed by immunoblotting.

The following antibodies were used: anti-pPAK1 (Thr423, Cell Signaling Technology, 1:1000, RRID:AB_330220), anti-PAK1 (Cell Signaling Technology, 1:1000, RRID:AB_330222), anti-pLIMK1 (Thr507/508, EMD Millipore, 1:1000, RRID:AB_568901), anti-LIMK1 (Cell Signaling Technology, 1:1000, RRID:AB_2281332), anti-pCofilin (Ser3, Cell Signaling Technology, 1:1000, RRID:AB_ 330238), anti-cofilin (Cell Signaling Technology, 1:1000, RRID:AB_10622000), anti-pSSH1L (Ser978, ECM Biosciences, 1:500, RRID:AB_10553849), anti-SSH1 (Cell Signaling Technology, 1:1000, RRID: AB_2798263), anti-Rac1 (Cell Biolabs, 1:6000), and anti-β-actin (Sigma-Aldrich, 1:6000, RRID:AB_ 476693).

## Rac1 activation assay

The Rac1 activation assay was performed using Cell Biolabs Rac1 Activity Assay Kit (#STA-401–1). Briefly, cells or tissue were lysed as described above and the active form of Rac1 (GTP-Rac1) was selectively pulled down from the lysate with p21-binding domain (PBD) of PAK agarose beads. Subsequently, the precipitated GTP-Rac1 was detected by Western blot analysis as described above. Each assay also consisted of lysates that were loaded with GTPγS or GDP as positive and negative controls respectively. A separate Western blot was run to evaluate total levels of Rac1 in each lysate.

## F/G-actin ratio

F/G-actin ratio was assessed as previously described *Pyronneau et al. (2017)*. Briefly, cells were lysed in cold lysis buffer [10 mM $K_2PO_4$, 100 mM NaF, 50 mM KCl, 2 mM $MgCl_2$, 1 mM EGTA, 0.2 mM DTT, 0.5% Triton-X 100, 1 mM sucrose (pH 7.0)] and centrifuged at 15,000 $x$ $g$ for 30 min. Separation of F-actin and G-actin was achieved in that F-actin is insoluble (pellet) in this buffer, whereas G-actin is soluble (supernatant). The G-actin supernatant was transferred to a fresh tube and the F-actin pellet was resuspended in lysis buffer plus an equal volume of a second buffer [1.5 mM guanidine hydrochloride, 1 mM sodium acetate, 1 mM $CaCl_2$, 1 mM ATP, 20 mM tris-HCl (pH 7.5)] and then incubated on ice for one hour with gentle mixing every 15 min to convert F-actin into soluble G-actin. Samples were centrifuged at 15,000 $x$ $g$ for 30 min and the supernatant (containing F-actin which was converted to G-actin) was transferred to a fresh tube. F-actin and G-actin samples were loaded with equal volumes and analyzed via Western blot. Latrunculin A (5 µM, 2 hr), a potent actin

polymerization inhibitor, and jaspakinolide (5 μM, 2 hr), an inducer of actin polymerization, were used as internal controls for the assay.

## Immunocytochemistry

Immunocytochemistry was performed as previously described *Pitcher et al. (2014)*. Cells were washed in PBS, fixed in 2% paraformaldehyde (PFA) for 10 min at room temperature and 4% PFA at 4°C for 20 min, and permeabilized with 0.1% Triton-X 100 for five minutes. Blocking was performed with 5% normal goat serum for 30 min. The following primary and secondary antibodies were used: anti-MAP2 (Millipore, 1:1000, RRID:AB_91939) and goat anti-rabbit Alexa Fluor 568 (Invitrogen, 1:250, RRID:AB_143157). Cells were counterstained with both Hoechst (Invitrogen, 1:10,000) and phalloidin Alexa Fluor 488 (Invitrogen, 1:400). After staining, coverslips were rinsed in $H_2O$ and mounted using ProLong Gold Antifade mounting media (Invitrogen).

## DiOlistic labeling of brain slices

Rats were sacrificed following the end of behavioral studies. Brains were rapidly removed, rinsed in $H_2O$, fixed in 4% PFA for 1 hr, and washed 3 times in 1X PBS. After fixation, serial coronal slices were sectioned via vibratome at a thickness of 150 μm and placed in tissue culture plates until further processing. DiOlistic labeling was performed according to published techniques (*Seabold et al., 2010*). 300 mg of tungsten beads (Bio-Rad, Hercules, CA) were suspended in 99.5% pure methylene chloride (Fisher Scientific) and sonicated in a water bath for 1 hr. Crystalized DiI (13.5 mg; Invitrogen) was dissolved in methylene chloride and protected from light. Following sonication, 100 μL of the tungsten bead solution was placed on a glass slide and 100 μL of DiI solution titrated on top, which was slowly mixed using a pipette tip and allowed to dry. A razor blade was used to collect the dry bead/dye mixture onto weigh paper, placed into a 15 mL conical tube with 3 mL ddH2O, and sonicated in a water bath for 20 min. The bead/dye mixture was drawn into Tezfel tubing coated with polyvinylpyrrolidone (Sigma-Aldrich, St. Louis, MO) and dried using nitrogen gas for 1 hr. Once dry, tubing was cut into 13 mm cartridges and loaded into the Helios Gen Gun (Bio-Rad). Helium gas flow was adjusted to 120 PSI and bullets were delivered to slices through 3 μm pore filter paper. Slices were quickly washed 3 times with 1X PBS and stored overnight at 4°C to allow diffusion of the dye. The following day, slices were mounted using ProLong Gold Antifade (Invitrogen), coverslipped, and stored at 4°C until imaging.

## Dendritic spine analysis

For in vitro experiments, neurons were cultured for 21 days, then fixed and stained for MAP2 (Alexa 568) and counterstained with phalloidin (Alexa 488) or infected with GFP-tagged lentiviral particles, as described above. Images were acquired with the Olympus FLUOVIEW FV3000 confocal microscope equipped with a 100x silicone oil-immersion objective at 2x electronic zoom and taken at 0.5 μm Z steps. Neurolucida 360 software (MBF Bioscience) was utilized to automatically quantify dendritic spines, as well as classify them into their respective morphologies based on established parameters (*Rodriguez et al., 2008*). For each experiment, four dendrites, at least 100–150 μm in length, from each coverslip were analyzed and a total of three coverslips were imaged for each condition. Each coverslip was averaged as a single data point and the experiment was repeated across three separate neuronal dissections.

For in vivo studies, dendrites in layer II/III pyramidal neurons from the prelimbic (PrL) region of the mPFC were imaged using the Olympus FLUOVIEW FV3000 using a 100x silicone oil-immersion objective at 2x electronic zoom and taken at 0.15 μm Z-steps. Neurolucida 360 software was used as mentioned above. Eight dendrites, at least 100–150 μm in length, were analyzed from eight separate neurons and averaged together as a single data point per animal.

## Tyramide stain and amplification

The hemisphere contralateral to the one used for DiOlistic staining was fixed in 4% PFA for 24 hr, moved to 70% ethanol, and then paraffin embedded and sectioned at 5 μm for immunohistochemical analysis. Tissue was sequential dual-stained with pPAK1 (Thr423, Invitrogen, 1:25, RRID:AB_2554427) and the neuronal marker NeuN (Cell Signaling, 1:400, RRID:AB_2651140). After rehydration, antigen retrieval was performed in a citrate buffer (Life Technologies) at 95°C for 20 min

followed by quenching of endogenous peroxidase activity with hydrogen peroxide and methanol for 30 min at room temperature. Tissue was blocked with 10% normal goat serum before incubation with primary antibody overnight at 4°C in a humidity chamber. Secondary antibody (goat anti-rabbit poly-HRP) incubation was performed for 1 hr at room temperature, then amplified with Alexa Fluor conjugated tyramide (Invitrogen, 488 for pPAK1, 555 for NeuN). Tissue was counterstained with Hoescht (Invitrogen, 1:10,000, 5 min), washed in $H_2O$, and then mounted with ProLong Gold Anti-fade mounting media.

## Multispectral imaging

Fluorescence microscopy coupled with multispectral image analysis was performed to identify NeuN + neurons within layer II/III of the PrL region in the mPFC. A spectral library was built to remove autofluorescence and separate out individual fluorescence spectra using Nuance software. The average optical density (OD) per area for pPAK1 was measured within each individual NeuN+ neuron. Two sections per animal were imaged, with three separate pictures taken per section and then averaged into a single data point per animal.

## Materials

Cell culture media was purchased from Invitrogen. Pertussis toxin (PTX) and AMD3100 were from Sigma Aldrich, NSC23766 was from Tocris Bioscience, Latrunculin A and Jaspakinolide were from Santa Cruz Biotechnology, and recombinant CXCL12 from Peprotech. Cannulas were custom-made and purchased from Plastics One. pCMVR8.74 (Addgene plasmid #22036, RRID:Addgene_22036) and pMD2.G (Addgene plasmid #12259, RRID:Addgene_12259) were gifts from Didier Trono.

## Experimental design and statistical analysis

The number of animals used for behavioral and in vivo studies was calculated by power analysis using PS: Power and Sample Size Calculation version 3.1.6 (Vanderbilt University). Sample sizes were calculated using an independent measures study design with $\alpha = 0.05$ and power = 0.8. Male WT or HIV-Tg F344 rats (approximately 4 months old) were randomly assigned to groups for behavioral and in vivo studies. In vivo data collection and analysis was conducted blinded to genotype and treatment. All in vitro experiments were performed with at least N = 3 biological replicates, and each biological replicate used neurons derived from a different litter of E17 rats. Outliers were neither encountered nor removed from any analysis. Data are represented as mean ± SEM for in vitro and in vivo experiments. The number of trials to criterion for each behavioral task was analyzed using a two-tailed Student's t-test or one-way ANOVA followed by Tukey post hoc. Dendritic spine density and morphology were analyzed using a two-tailed Student's t-test or one-way ANOVA followed by Dunnett post hoc if comparing experimental groups to a control group or Tukey post-hoc if comparing all groups against each other. The correlation between spine density and trials to criterion was calculated as Pearson's *r*. The rate for reaching criterion was assessed by linear regression analysis to determine if the slopes of each line were equal. For multispectral imaging, a two-tailed Student's t-test was used. For the remaining in vitro studies, statistical significance was determined by two-tailed Student's t-test or one-way ANOVA followed by Dunnett post hoc. A $p<0.05$ was considered statistically significant. P and n are reported in the figure legends; for correlation analysis, r values are also reported in the text. Additional information regarding statistical analyses and raw data are provided in the source data files for each figure. All statistical analysis was performed with GraphPad Prism, version 8.0 (GraphPad Software, RRID:SCR_015382).

## Acknowledgements

The authors thank Renato Brandimarti for assistance with viral particle production, David Sulzer (Columbia University) and members of the Meucci laboratory for discussion and critical reading of the manuscript, particularly Bradley Nash for support with editing and statistics.

## Additional information

### Funding

| Funder | Grant reference number | Author |
| --- | --- | --- |
| National Institute on Drug Abuse | DA015014 | Olimpia Meucci |
| National Institute on Drug Abuse | DA032444 | Olimpia Meucci |
| National Institute on Drug Abuse | DA040519 | Olimpia Meucci |
| National Institute of Mental Health | MH078795 | Lindsay K Festa |
| Natural Sciences and Engineering Research Council of Canada | | Stan Floresco |

The funders had no role in study design, data collection and interpretation, or the decision to submit the work for publication.

### Author contributions

Lindsay K Festa, Conceptualization, Data curation, Formal analysis, Investigation, Methodology; Elena Irollo, Formal analysis, Investigation; Brian J Platt, Yuzen Tian, Investigation, Methodology; Stan Floresco, Software, Methodology; Olimpia Meucci, Conceptualization, Resources, Formal analysis, Supervision, Funding acquisition, Investigation, Visualization, Methodology, Project administration

### Author ORCIDs

Lindsay K Festa https://orcid.org/0000-0002-5419-9532
Olimpia Meucci https://orcid.org/0000-0001-8333-4804

### Ethics

Animal experimentation: This study was performed using rats singly housed in isolation in our Association for Assessment and Accreditation of Laboratory Animal Care-accredited barrier facilities in accordance with the National Institutes of Health guidelines and institutional approval by the Institutional Animal Care and Use Committee (Drexel University protocol #20733 and 20732).

### Decision letter and Author response

Decision letter https://doi.org/10.7554/eLife.49717.sa1
Author response https://doi.org/10.7554/eLife.49717.sa2

## Additional files

### Supplementary files

• Transparent reporting form

### Data availability

All data presented in this manuscript are available as source data files.

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
