## [Decision Letter]

**Acceptance summary:**

Cognitive deficits are a serious complication associated with AIDS even in treated subjects. Festa and colleagues report that activation of CXCL12/CXCR4 signaling in prefrontal cortex of HIV-Tg rat, an animal model of HAND and provide evidence that the pathological deficits in dendritic spine density and maturation can be reversed by infusion of CXCL12 through the Rac1/PAK actin polymerization pathway. The authors report an interesting set of findings identify a new mechanism and open up important new directions of investigation.

**Decision letter after peer review:**

Thank you for submitting your article "CXCL12-induced rescue of cortical dendritic spines and cognitive flexibility" for consideration by *eLife* and your patience. Your article has been reviewed by two peer reviewers, and the evaluation has been overseen by a Reviewing Editor and Jonathan Cooper as the Senior Editor. The following individual involved in review of your submission has agreed to reveal their identity: Norman Haugey (Reviewer #2).

The reviewers have discussed the reviews with one another and the Reviewing Editor has drafted this decision to help you prepare a revised submission.

Summary:

The manuscript by Festa and colleagues reports activation of CXCL12/CXCR4 signaling in prefrontal cortex of HIV-Tg rat, an animal model of HAND. Moreover, they provide evidence that the pathological deficits in dendritic spine density and maturation can be reversed by infusion of CXCL12 through the Rac1/PAK actin polymerization pathway.

Specifically, the authors showed that HIV-Tg rat showed 1) impaired attentional set-shifting behavior, which is correlated with the loss of spine density in the mPFC 2) CXCL12 treatment through ICV infusion rescued the deficit in spine density and attentional set-shifting behavior and 3) CXCL12-induced rescue effect was mediated through the CXCR4/Rac1/PAK actin polymerization pathway.

Cognitive deficits are a serious complication associated with AIDS even in treated subjects. The authors report an interesting set of findings that start to identify a new mechanism. However, in vivo validation and evidence of this CXCL12/CXCR4/Rac1/PAK signaling mechanism would significantly strengthen this study and its impact. Moreover, a few key points and questions need to be addressed as outlined below.

Essential revisions:

1) Is there a deficit in CXCL12/CXCR4/Rac1/PAK signaling in the HIV-Tg model? Most of the studies of signaling pathways were carried out with cultured WT neurons. It would be important to characterize the signaling changes in PFC of the HIV-Tg model.

2) CXCR4-independent effects of CXCL12 should be considered. Chemokines are known to have multiple targets. For example, CXCL12 can activate receptors such as CXCR7. Please address in Discussion.

Additional comments:

1) Is the result in Figure 1A sufficiently powered? It appears that the HIV-Tg result is higher than WT.

2) The results in Figure 5A, 5B, 5C and Figure 6C suggest that the pLIMK1/LIMK1 findings are not consistent.

3) Thin spine density in Figure 8D (WT) is similar with Figure 9D (HIV-Tg), which is not consistent with Figure 3D.

4) The method for labelling spines in vivo does not seem to be included in the paper

Representative images of in vivo spine density and morphology for some of the in vivo pathological and behavioral reversals would be useful for completeness.

5) It is interesting that the most robust effect of CXCL12 appears to be on thin spines with a reciprocal reduction in stubby spines. These findings are entirely consistent with their data on cytoskeletal stabilization, and suggest that CXCL12 promotes the elongation of stubby spines. It is somewhat curious that the behavioral effects do not appear to correlate with the increased presence of mushroom shaped spines, as these presumably would be synaptically connected. It is possible this effect is due to the timing of CXCL12 infusions, behavioral testing and sacrifice date. i.e. it is possible that the increase of mushroom shaped spines occurred earlier during the infusion regimen and subsequently downregulated after the behavioral task was mastered, or it is possible that the presence of mushroom shaped spines does not accurately reflect synaptic connectivity. PSD95 staining or Western would address the curiosity. If the authors do not have sufficient tissue left over (even contralateral side may be sufficient based on their data), a further discussion of this phenomenon would be sufficient.

---

## [Author Response]

Essential revisions:1) Is there a deficit in CXCL12/CXCR4/Rac1/PAK signaling in the HIV-Tg model? Most of the studies of signaling pathways were carried out with cultured WT neurons. It would be important to characterize the signaling changes in PFC of the HIV-Tg model.

This issue is addressed in Figure 7, which has been updated to clearly show CXCL12-induced activation of PAK in the PFC of WT and HIV-Tg rats. Additionally, new related studies are included in Figure 7—figure supplement 1. The short answer to the reviewer’s question is *No* – i.e. the ability of the chemokine to activate Rac1/PAK signaling remains intact in the HIV-Tg model.

More specifically, though the level of PAK phosphorylation in PFC neurons of control (i.e. vehicle treated) HIV-Tg rats is lower than in the corresponding WT group, CXCL12 effectively induces PAK phosphorylation in the HIV-Tg brain and this stimulation is as pronounced as that observed in WT rats (Figure 7A). When compared to the basal value in their respective controls, the magnitude of pPAK increase is comparable in the two groups (average signal per area D = 37 in Tg rats and 36 in WT rats; Figure 7A bottom graph). As predicted from the robust CXCL12-induced PAK stimulation in both genotypes, we did not find significative difference in the expression of total Rac1 protein in HIV-Tg PFC tissue versus WT (Figure 7—figure supplement 1).However, in line with the above pPAK results under basal conditions, pulldown studies from PFC of untreated rats show reduced levels of active Rac1 in HIV-Tg compared to untreated WT (Figure 7—figure supplement 1) – suggesting a reduced basal tone of PAK activation in the HIV-Tg model.

Overall, these studies indicate that the CXCL12/CXCR4/Rac1/PAK pathway is fully functional in the HIV-Tg model – a conclusion further supported by the finding that the ability of CXCL12 to rescue spine deficits in the HIV-Tg model is completely blocked by the Rac1 inhibitor (Figure 9).

2) CXCR4-independent effects of CXCL12 should be considered. Chemokines are known to have multiple targets. For example, CXCL12 can activate receptors such as CXCR7. Please address in Discussion.

We recognize the contribution of CXCR7 to other CXCL12 functions and, as suggested by the reviewer, we have discussed this point in the revised manuscript. However, evidence pointing to a main role of CXCR4 in these effects comes not only from the signaling studies presented in this manuscript (Figure 4), but also from previous findings concerning the role of CXCL12/CXCR4 in excitatory neurotransmission and its regulation by endogenous and exogenous factors (e.g. Sengupta et al., 2009, Nicolai et al., 2010, Pitcher et al., 2014, Nash et al., 2019). Additionally, CXCR7 expression on the plasma membrane of differentiated neurons is thought to be modest in the adult brain (e.g. Shimizu et al., 2011). This receptor is certainly critical during development, as it regulates migration of neural precursors and CXCR4 signaling. However, it does not signal via the canonical G-protein coupling and appears less relevant in this specific context.

Additional comments:1) Is the result in Figure 1A sufficiently powered? It appears that the HIV-Tg result is higher than WT.

The reviewer is correct – we performed additional statistical analysis and revised both Figure 1 and the text. Thank you for pointing this out.

2) The results in Figure 5A, 5B, 5C and Figure 6C suggest that the pLIMK1/LIMK1 findings are not consistent.

The experiments reported in these figures were performed in primary neurons and conducted over a period of several months. Thus, some variability in the time course is to be expected as these studies come from 12 different cultures (n=3 for each individual set of experiment) and multiple rat embryos. That said, CXCL12 consistently stimulated PAK activation, with peaks at 5-15 minutes. LIMK1 phosphorylation is downstream PAK and was also consistently activated during the first 30-60 minutes, as one would expect. While there is biological variability due to the use of primary tissue over an extended period of time and the complexity of the biochemical process in question, we do typically see subsequent stimulation of the downstream targets within the PAK/LIMK1/cofilin pathway.

3) Thin spine density in Figure 8D (WT) is similar with Figure 9D (HIV-Tg), which is not consistent with Figure 3D.

In addition to intrinsic biological variability, this may be explained by minor differences related to the source of animals, which changed toward the end of this project. All the experiments utilized the transgenic line generated by Bryant’s group at the University of Maryland in 2001 (Reid et al., 2001). However, until recently breeding of the HIV-Tg was licensed to a commercial vendor, i.e. Envigo. In the summer of 2017 Envigo’s license was not renewed and they returned the colony to the University of Maryland. At that point, we started purchasing the animals from the University of Maryland directly. Hence, while the experiments in Figures 1, 2, 3, 6, 7, 8 were entirely performed with rats from Envigo, for Figure 9 we used a first set of animals (n=12) from Envigo but the rest (n=24) were provided by the University of Maryland animal facility. Importantly, for both cohort of animals we run the four different groups (Vehicle, CXCL12, NSC, NSC+CXCL12) at the same time. Despite the modest change in the number of thin spines compared to the past (possibly due to slight difference in animal age, housing or environment), the effect of CXCL12 and NSC were highly reproducible within the different cohorts – which further increased our confidence in the results. Indeed, independently of the thin spine density under basal conditions, CXCL12’s effect was clear, and it was consistently inhibited by NSC. Furthermore, the changes in thin spine density nicely correlated with cognitive performance as in the previous studies.We have clarified the issue about the animal source in the revised manuscript.

4) The method for labelling spines in vivo does not seem to be included in the paperRepresentative images of in vivo spine density and morphology for some of the in vivo pathological and behavioral reversals would be useful for completeness.

The method is included in the section titled “*Diolistic labeling of brain slices*”. As requested, we have added representative images in Figures 2A, 3C and Figure 3—figure supplement 4.

5) It is interesting that the most robust effect of CXCL12 appears to be on thin spines with a reciprocal reduction in stubby spines. These findings are entirely consistent with their data on cytoskeletal stabilization, and suggest that CXCL12 promotes the elongation of stubby spines. It is somewhat curious that the behavioral effects do not appear to correlate with the increased presence of mushroom shaped spines, as these presumably would be synaptically connected. It is possible this effect is due to the timing of CXCL12 infusions, behavioral testing and sacrifice date. i.e. it is possible that the increase of mushroom shaped spines occurred earlier during the infusion regimen and subsequently downregulated after the behavioral task was mastered, or it is possible that the presence of mushroom shaped spines does not accurately reflect synaptic connectivity. PSD95 staining or Western would address the curiosity. If the authors do not have sufficient tissue left over (even contralateral side may be sufficient based on their data), a further discussion of this phenomenon would be sufficient.

We thank the reviewer for the comment and agree on the complexity of the issue. Live imaging studies aiming at further exploring the relationship between the reciprocal changes in thin and stubby spines will help address this question. The idea that CXCL12 promotes elongation of stubby spines is attractive, but it is also possible that CXCL12 stabilizes thin spines by reducing their turnover rate. Spine changes may also result from E/I circuit adaptations. We too were intrigued by potential changes in PSD95 and have been investigating this issue recently. So far, we have not observed modifications in overall PSD95 expression in cultured neurons exposed to CXCL12, but we are currently investigating possible changes at the synaptic level specifically (both in cultured neurons and brain tissue slices). In regard to mushroom spines, they were not significantly affected by CXCL12. This is not that surprising as mushroom spines are the most mature and stable type of spines, typically implicated in long term memory. Thin spines on the other hand are thought to be critical for cognitive flexibility. We modified the Discussion to cover these valid points.